

# The Polar Amplification Model Intercomparison Project (PAMIP) contribution to CMIP6: investigating the causes and consequences of polar amplification

Doug M. Smith[1], James A. Screen[2], Clara Deser[3], Judah Cohen[4], John C. Fyfe[5], Javier García-Serrano[6,7], Thomas Jung[8,9], Vladimir Kattsov[10], Daniela Matei[11], Rym Msadek[12], Yannick Peings[13], Michael Sigmond[5], Jinro Ukita[14], Jin-Ho Yoon[15], Xiangdong Zhang[16]

[1]Met Office Hadley Centre, Exeter EX1 3PB, UK
[2]College of Engineering, Mathematics and Physical Sciences, University of Exeter, Exeter EX4 4QE, UK
[3]Climate and Global Dynamics, National Center for Atmospheric Research, Boulder CO 80305, USA
[4]Atmospheric and Environmental Research, Lexington, MA, USA
[5]Canadian Centre for Climate Modelling and Analysis, Environment and Climate Change Canada, Victoria, British Columbia V8W 2Y2, Canada
[6]Barcelona Supercomputing Center (BSC), Barcelona, Spain
[7]Group of Meteorology, Universitat de Barcelona, Barcelona, Spain
[8]Alfred Wegener Institute, Helmholtz Centre for Polar and Marine Research, Bremerhaven, Germany
[9]University of Bremen, Bremen, Germany
[10]Voeikov Main Geophysical Observatory, Roshydromet, Russia
[11]Max-Plank-Institut für Meteorologie, Hamburg, Germany
[12]CERFACS/CNRS, UMR 5318, Toulouse, France
[13]Department of Earth System Science, University of California, Irvine, Irvine, California, USA
[14]Institute of Science and Technology, Niigata University, Niigata, Japan
[15]Gwangju Institute of Science and Technology, School of Earth Sciences and Environmental Engineering, Gwangju, South Korea
[16]International Arctic Research Center, University of Alaska Fairbanks, Fairbanks AK 9775, USA

*Correspondence to*: Doug M. Smith (doug.smith@metoffice.gov.uk)

**Abstract.**

Polar amplification – the phenomenon that external radiative forcing produces a larger change in surface temperature at high latitudes than the global average – is a key aspect of anthropogenic climate change, but its causes and consequences are not fully understood. The Polar Amplification Model Intercomparison Project (PAMIP) contribution to the Sixth Coupled Model Intercomparison Project (CMIP6, Eyring et al. 2016) seeks to improve our understanding of this phenomenon through a coordinated set of numerical model experiments documented here. In particular, PAMIP will address the following primary questions:

1. What are the relative roles of local sea ice and remote sea surface temperature changes in driving polar amplification?

2. How does the global climate system respond to changes in Arctic and Antarctic sea ice?

These issues will be addressed with multi-model simulations that are forced with different combinations of sea ice and/or sea surface temperatures representing present day, pre-industrial and future conditions. The use of three time periods allows the signals of interest to be diagnosed in multiple ways. Lower priority tier experiments are proposed to investigate additional




aspects and provide further understanding of the physical processes. These experiments will address the following specific questions: What role does ocean-atmosphere coupling play in the response to sea ice? How and why does the atmospheric response to Arctic sea ice depend on the pattern of sea ice forcing? How and why does the atmospheric response to Arctic sea ice depend on the model background state? What are the roles of local sea ice and remote sea surface temperature in polar

amplification, and the response to sea ice, over the recent period since 1979? How does the response to sea ice evolve on decadal and longer timescales?

A key goal of PAMIP is to determine the real world situation using imperfect climate models. Although the experiments proposed here form a coordinated set, we anticipate a large spread across models. However, this spread will be exploited by

seeking "emergent constraints" in which model uncertainty may be reduced by using an observable quantity that physically explains the inter-model spread. In summary, PAMIP will improve our understanding of the physical processes that drive polar amplification and its global climate impacts, thereby reducing the uncertainties in future projections and predictions of climate change and variability.

## 1 Introduction

Polar amplification refers to the phenomenon in which zonally-averaged surface temperature changes in response to climate forcings are larger at high latitudes than the global average. Polar amplification, especially in the Arctic, is a robust feature of global climate model simulations of recent decades (Bindoff et al. 2013) and future projections driven by anthropogenic emissions of carbon dioxide (Figure1, Collins et al. 2013). Polar amplification over both poles is also seen in simulations of paleo-climate periods driven by solar or natural carbon cycle perturbations (Masson-Delmotte et al. 2013).

Observations over recent decades (Figure 2) suggest that Arctic amplification is already occurring: recent temperature trends in the Arctic are about twice the global average (Serreze et al. 2009; Screen and Simmonds 2010; Cowtan and Way 2013), and Arctic sea ice extent has declined at an average rate of around 4% per decade annually and more than 10% per decade during the summer (Vaughan et al. 2013). Climate model simulations of the Arctic are broadly consistent with the observations (Figure

2). However, there is a large inter-model spread in temperature trends (Bindoff et al. 2013), the observed rate of sea ice loss is larger than most model simulations (Stroeve et al. 2012), and the driving mechanisms are not well understood (discussed further below). Antarctic amplification has not yet been observed (Figure 2). Indeed, Antarctic sea ice extent has increased slightly over recent decades (Vaughan et al. 2013) in contrast to most model simulations (Bindoff et al. 2013), and understanding recent trends represents a key challenge (Turner and Comiso 2017). Nevertheless, Antarctic amplification is

expected in the future in response to further increases in greenhouse gases, but is likely to be delayed relative to the Arctic due to strong heat uptake in the Southern Ocean (Collins et al. 2013; Armour et al., 2016). There is mounting evidence that polar

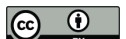



amplification will affect the global climate system by altering the atmosphere and ocean circulations, but the precise details and physical mechanisms are poorly understood (discussed further below).

The Polar Amplification Model Intercomparison Project (PAMIP) will investigate the causes and global consequences of polar
amplification, through creation and analysis of an unprecedented set of coordinated multi-model experiments and strengthened international collaboration. The broad scientific objectives are:

- Provide new multi-model estimates of the global climate response to Arctic and Antarctic sea ice changes.
- Determine the robustness of the responses between different models and the physical reasons for differences.
- Improve our physical understanding of the mechanisms causing polar amplification and its global impacts.
- Harness increased process understanding and new multi-model ensembles to constrain projections of future climate change in the polar regions and associated global climate impacts.

PAMIP will directly contribute to the World Climate Research Programme (WCRP) Grand Challenges on Near-term Climate Prediction, Melting Ice and Global Consequences, and Weather and Climate Extremes, and addresses all three of the Sixth
Coupled Model Intercomparison Project (CMIP6, Eyring et al. 2016) scientific questions:

1.  How does the Earth system respond to forcing? This will be addressed through coordinated multi-model experiments to understand the causes and consequences of polar amplification.
2.  What are the origins and consequences of systematic model biases? Specific experiments are proposed to investigate the role of model biases in the atmospheric response to sea ice.
3.  How can we assess future climate changes given climate variability, predictability and uncertainties in scenarios? Analysis of PAMIP experiments will focus on process understanding in order to constrain future projections.

This paper describes the motivation for PAMIP, and documents the proposed model experiments and suggested analysis procedure. An overview of the causes and consequences of polar amplification is given in Sections 2 and 3 before outlining
the need for coordinated model experiments in Section 4. The proposed PAMIP experiments and analysis are documented in Section 5 and the data request is described in Section 6. Interactions with other MIPs are discussed in Section 7. A summary is provided in Section 8, and data availability is described in Section 9. Details of the forcing data are given in Appendix A, and technical details for running the experiments are given in Appendix B.

## 2 Causes of polar amplification

Several feedback mechanisms contribute to polar amplification, and operate at both low and high latitudes (Taylor et al. 2013; Pithan and Mauritsen 2014). The most well established is the surface albedo feedback at high latitudes (Manabe and Stouffer 1994; Hall 2004) in which melting of highly reflective sea ice and snow regions results in increased absorption of solar



radiation which amplifies the warming. However, lapse rate and Planck feedbacks play a larger role in climate model simulations of Arctic amplification than the surface albedo feedback (Pithan and Mauritsen 2014). Lapse rate feedback is negative at lower latitudes where the upper troposphere is heated by latent heat released by rising air parcels, but becomes positive at high latitudes where the more stable atmosphere restricts local surface-driven warming to low altitudes. Hence, 5 lapse rate feedback directly drives polar amplification by acting to reduce the warming at low latitudes and increase the warming at high latitudes. Planck feedback is negative everywhere and opposes global warming by emission of long wave radiation. However, it operates more strongly at warmer lower latitudes and therefore contributes to polar amplification. Other feedbacks are also potentially important in controlling temperature trends in polar regions, including water vapour (Graversen and Wang 2009), cloud (Vavrus 2004), and changes in heat transport in the atmosphere (Manabe and Wetherald 1980) and 10 ocean (Khodri et al 2001; Holland and Bitz 2003; Spielhagen et al. 2011). Although some of these may operate more strongly at lower latitudes thereby opposing polar amplification (Pithan and Mauritsen 2014) they are important in controlling the overall temperature trends and hence the magnitude of polar amplification.

Factors other than anthropogenic increases in greenhouse gases (GHGs) modulated by the feedbacks outlined above have also 15 contributed to recent temperature trends, potentially enhancing or inhibiting the observed rates of polar amplification. Arctic warming rates over the last century have likely been modulated by changes in solar radiation, volcanic eruptions and anthropogenic aerosol emissions (Overpeck et al. 1997; Fyfe et al. 2013; Acosta Navarro et al. 2016; Gagné et al. 2017), and by decadal timescale variations in Atlantic and Pacific sea surface temperatures (Chylek et al. 2009; Ding et al. 2014, 2017; Tokinaga et al. 2017), referred to here as Pacific Decadal Variability (PDV) and Atlantic Multidecadal Variability (AMV). 20 Recent Antarctic temperature and sea ice trends (Figure 2) have likely been strongly influenced by changes in atmospheric circulation (Turner et al 2015; Raphael et al., 2015; Jones et al. 2016b), notably an increase in the southern annular mode (SAM) and a deepening of the Amundsen Sea Low (ASL). The SAM increase, especially during austral summer, has been linked to ozone depletion (Thompson and Solomon 2002), but its role in driving warming of the Antarctic Peninsula, which peaks in winter and spring, is unclear (Smith and Polvani 2017). The deepening of the ASL has likely been influenced by both 25 PDV (Purich et al. 2016; Schneider et al. 2015; Schneider and Deser 2017) and AMV (Li et al. 2014). Freshening of Antarctic surface waters from melting ice shelves may also have influenced recent Antarctic sea ice and temperature trends (Bintanja et al. 2013), though the magnitude of this effect is uncertain (Swart and Fyfe 2013).

## 3 Consequences of polar amplification

Polar amplification will affect the melting of polar ice sheets and hence sea level rise, and the rate of carbon uptake in the polar 30 regions. These impacts are investigated by the Ice Sheet Model Intercomparison Project (Nowicki et al. 2016) and the Coupled Climate–Carbon Cycle Model Intercomparison Project (Jones et al. 2016a), respectively. PAMIP, described here, will focus on the impacts of sea ice changes on the global climate system through changes in the atmosphere and ocean circulation. This



is an area of intensive scientific interest and debate, as summarised in several recent reviews (Cohen et al. 2014; Vihma 2014; Walsh 2014; Barnes and Screen 2015; Overland et al. 2015, Screen et al. 2018). A number of hypothesised consequences of a warming Arctic have been proposed based on observations, including changes in the behaviour of the polar Jet Stream (e.g. Francis and Vavrus 2012), that could potentially give rise to more persistent and extreme weather events. Arctic warming has

also been proposed as a cause of decadal cooling trends over Eurasia (Liu et al. 2012; Mori et al. 2014; Kretschmer et al. 2017), in what has been referred to as the Warm-Arctic Cold-Continents pattern (Overland et al. 2011; Cohen et al. 2013). However, determining causality solely from observations is an intractable problem. For this reason, model experiments with reduced sea ice have been extensively used - but to date, with little coordination between modelling groups - in an attempt to better isolate the response to sea ice loss and understand the causal mechanisms. Such experiments tend to broadly agree on the local

thermodynamic response but diverge considerably on the dynamical response. A key area of uncertainty is the atmospheric circulation changes in responses to Arctic warming. By definition, polar amplification will reduce the equator to pole surface temperature gradient, potentially weakening the mid-latitude westerly winds and promoting a negative Arctic Oscillation (AO) or North Atlantic Oscillation (NAO). However, this dynamic response may be opposed by a local thermodynamic low-pressure response to Arctic warming that acts to strengthen the mid-latitude westerlies (Smith et al. 2017), and the overall response is

unclear (Deser et al., 2015; Shepherd 2016). Since the remote consequences of polar amplification are to a large extent governed by changes in the atmospheric circulation, it is of critical importance to attempt to constrain the circulation response to polar amplification through collaborative modelling activities. The response to Antarctic sea ice has received far less attention. Some studies simulate an equatorward shift of the mid-latitude tropospheric jet in response to reduced Antarctic sea ice (Raphael et al. 2011; Bader et al. 2013; Smith et al. 2017), but it is unclear whether this relationship will continue to hold

in future as the sea ice retreats (Kidston et al. 2011). Polar amplification will also drive changes in the ocean, potentially giving rise to global climate impacts. For example, reduced Arctic sea ice may weaken the Atlantic Meridional Overturning Circulation (Sévellec et al. 2017; Suo et al. 2017); and increased warming of the Northern Hemisphere relative to the Southern Hemisphere may shift the inter-tropical convergence zone (Chiang and Bitz 2005) affecting Sahel rainfall and tropical storm activity (Smith et al. 2017), and Californian drought (Cvijanovic et al. 2017). However, the extent to which the latter impacts

are mitigated by changes in ocean heat transport convergence is uncertain (Tomas et al. 2016).

**4 The need for coordinated model experiments**

It is clear from the discussion in Sections 2 and 3 that both the causes and consequences of polar amplification are uncertain. CMIP6 (Eyring et al. 2016) provides an unprecedented opportunity to improve our understanding of climate change and variability in general, and several CMIP6 Model Intercomparison Projects (MIPs) will provide valuable information on the

causes and consequences of polar amplification in particular, as discussed in Section 7. PAMIP will complement the other CMIP6 MIPs by providing coordinated model experiments that are specifically designed to investigate the physical mechanisms driving polar amplification and the climate system's response to changes in sea ice. Improved understanding of



the physical processes gained through PAMIP will enable uncertainties in future polar amplification and associated climate impacts to be reduced.

A key uncertainty regarding the causes of polar amplification is the relative role of local processes that directly affect the surface energy budget and remote processes that affect the poleward heat transport. Local processes are likely to induce a response that is strongest near to the surface (Screen and Simmonds 2010), whereas changes in atmospheric heat transport may affect the mid-troposphere more strongly (Graverson et al. 2008). Observations of recent Arctic temperature trends show warming throughout the lower to mid-troposphere (Figure 3a) suggesting that both local and remote processes could be important, but assessing their relative roles is not possible from observations alone. Model experiments in which remote sea surface temperature (SST) and sea ice concentration (SIC) changes are imposed separately (Kumar et al. 2010; Screen et al. 2012; Perlwitz et al. 2015) enable the contributions from local and remote processes to be quantified (Figure 3b-d), and will be a core component of PAMIP.

It is also not possible to diagnose the climate system response to sea ice from observations alone. This is illustrated in Figure 4, which compares the winter mean sea level response to reduced Arctic sea ice inferred from lagged regression with the simulated response obtained in model experiments driven by changes in sea ice (Smith et al. 2017). Lagged regression shows a pattern that projects onto a negative NAO in both the observations and in atmosphere model experiments driven by observed SIC and SST (Figure 4a and b). These regressions imply a negative NAO response to reduced Arctic sea ice (e.g., Liu et al. 2012). However, the actual response to reduced Arctic sea ice determined from specific experiments using the same model is a weak positive NAO (Figure 4c). This suggests that the model response (Figure 4b) is driven by changes in SST rather than SIC. Hence, although statistical analysis can provide useful insights, the results can sometimes be misleading and need to be supported by dedicated model experiments. However, modelling studies currently simulate a full spectrum of NAO responses to reduced Arctic sea ice including negative NAO (Honda et al. 2009; Seierstad and Bader 2009; Mori et al. 2014; Kim et al. 2014; Deser et al. 2015; Nakamura et al. 2015), positive NAO (Singarayer et al. 2006; Strey et al. 2010; Orsolini et al. 2012; Rinke et al. 2013; Cassano et al. 2014; Screen et al. 2014), little response (Screen et al. 2013; Petrie et al. 2015; Blackport and Kushner 2016), and a response that depends on the details of the forcing (Alexander et al. 2004; Petoukhov and Semenov 2010; Peings and Magnusdottir 2014; Sun et al. 2015; Pedersen et al. 2016; Chen et al. 2016). There are many potential reasons for the different responses found in modelling studies, including:

- *Differences in the magnitude of the forcing.* Some studies have investigated the response to sea ice perturbations typical of the present day and near future (e.g., Chen et al. 2016; Smith et al. 2017), while others have investigated the impact of larger changes expected towards the end of the century (e.g,. Deser et al. 2016; Blackport and Kushner 2016). Furthermore, interpreting the impact of differences in the magnitude of the forcing is particularly difficult because the relationship could be non-linear (Petoukhov and Semenov 2010; Peings and Magnusdottir 2014; Semenov and Latif 2015; Chen et al. 2016).



- *Differences in the pattern of forcing.* Studies have demonstrated that the response is sensitive to the pattern of sea ice anomalies. For example, Sun et al. (2015) obtained opposite responses in the northern polar vortex to sea ice forcing from the Pacific and Atlantic sectors. Furthermore, the responses to regional sea ice anomalies do not add linearly (Screen 2017), complicating their interpretation.

- *How the forcing is applied.* Changes in sea ice can be imposed in different ways in coupled models, for example by nudging the model to the required state (e.g., Smith et al. 2017, McCusker et al. 2017), or by changing the fluxes of energy in order to melt some of the sea ice (e.g., Deser et al. 2016; Blackport and Kushner 2016). The impact of these different approaches is not clear, but they could potentially contribute to the spread of results.

- *Additional forcings.* Isolating the response to sea ice can be complicated if additional forcings are imposed. For example, greenhouse gas induced warming of the tropical troposphere tends to increase the equator to pole temperature gradient in the mid-troposphere and oppose the impact of sea ice (Deser et al., 2015; Blackport and Kushner 2017; Oudar et al. 2017).

- *Different models.* The response can be very sensitive to the model used. For example, Sun et al. (2015) obtained opposite responses in the winter polar vortex in identical forcing experiments with two different models, one with a well-resolved stratosphere and one without. García-Serrano et al. (2017) further discuss the diversity of potential Arctic-midlatitude linkages found in coupled models.

- *Atmosphere/ocean coupling.* Although many studies have used atmosphere-only models, changes in Arctic sea ice can influence sea surface temperatures (SSTs) surrounding the ice pack and also in remote regions, including the tropics (e.g., Deser et al. 2015; Tomas et al 2016; Smith et al. 2017). Coupled models are essential to simulate these effects, and may also amplify the winter mid-latitude wind response to Arctic sea ice (Deser et al. 2016).

- *Background state.* Identical experiments with the same model but with different background states induced by different SST biases can produce opposite NAO responses (Smith et al. 2017). Furthermore, responses may not be robust across experiments due to strong nonlinearities in the system, which can depend on the background state (Chen et al. 2016).

- *Land surface.* Snow cover can also influence the atmospheric circulation (Cohen and Entekhabi 1999; Gastineau et al., 2017), although models appear to underestimate the effects (Furtado et al. 2015). Differences in snow cover could therefore contribute to differences in modelled responses to sea ice.

- *Low signal-to-noise ratio.* The atmospheric response to Arctic sea ice simulated by models is typically small compared to internal variability so that a large ensemble of simulations is required to obtain robust signals (e.g., Screen et al., 2014; Mori et al. 2014). Some of the different responses reported in the literature could therefore arise from sampling errors. If the low signal-to-noise ratio in models is correct, then the response to Arctic sea ice could be overwhelmed by internal variability (McCusker et al. 2016). However, the signal-to-noise ratio in seasonal



forecasts of the NAO is too small in models (Eade et al. 2014; Scaife et al. 2014; Dunstone et al. 2016), suggesting that the magnitude of the simulated response to sea ice could also be too small.

PAMIP seeks to reduce these sources of differences since all simulations will follow the same experimental protocol, allowing
the different model responses to be better understood. Additional experiments will also focus on understanding the roles of coupling, the background state, and the pattern of forcing.

## 5 PAMIP experiments and analysis plan

Coordinated model experiments in PAMIP will address the following primary questions:

1. What are the relative roles of local sea ice and remote sea surface temperature changes in driving polar amplification?
2. How does the global climate system respond to changes in Arctic and Antarctic sea ice?

These questions will be answered by taking differences between numerical model simulations that are forced with different combinations of SST and/or SIC (Table 1) representing present day (pd), pre-industrial (pi) and future (fut, 2 degree warming) conditions. Pairs of simulations with the same SSTs but different SICs provide estimates both of the contribution of sea ice changes to polar amplification and of the climate response to sea ice changes. Pairs of simulations with the same SICs but
different SSTs provide estimates of the contribution of SST changes to polar amplification. The use of three periods allows the signals of interest to be diagnosed in multiple ways. Details of the forcing fields are given in Appendix A, and example SIC and SST forcing fields are shown in Figures 5 and 6 for the Arctic and Figures 7 and 8 for the Antarctic.

The Tier 1 experiments are atmosphere-only to minimise computational costs, and lower tier experiments investigate additional
aspects and provide further understanding of the physical processes. All experiments require a large ensemble to obtain robust results (Screen et al. 2014; Mori et al. 2014). The experiments are listed in Table 1, and further technical details are given in Appendix B. Suggested combinations for diagnosing the roles of SST and sea ice in polar amplification, and the climate response to sea ice, are given in Table 2. The experiments are grouped into six sets as follows:

1. *Atmosphere-only time slice.* What are the relative roles of local sea ice and remote sea surface temperature changes
25       in driving polar amplification, and how does the global climate system respond to changes in Arctic and Antarctic sea ice? This set contains all of the Tier 1 experiments, and provides a multi-model assessment of the primary scientific questions addressed by PAMIP. The difference between experiments 1.1 and 1.3 provides an estimate of the contribution of SST to the polar amplification between pre-industrial and present day conditions. The contribution of Arctic (Antarctic) sea ice to polar amplification, as well as the atmospheric response to Arctic (Antarctic) sea ice
30       changes between pre-industrial and present day conditions can be obtained by differencing experiments 1.1 and 1.5 (1.7). Similarly, the contribution of Arctic (Antarctic) sea ice to future polar amplification, as well as the atmospheric response to Arctic (Antarctic) sea ice changes between present day and future conditions can be obtained by



differencing experiments 1.1 and 1.6 (1.8). Tier 2 experiment 1.4 provides an estimate of the contribution of SST to future polar amplification, and 1.2 provides an additional estimate of the contribution of SST to past polar amplification. Further estimates can be obtained by differencing future and pre-industrial periods. Sensitivity to the magnitude of the forcing can also be investigated since differences in SST and SIC between future and pre-industrial conditions are much larger than between present day and future or pre-industrial conditions. Tier 3 experiments 1.9 and 1.10 enable the role of Arctic sea ice thickness changes to be assessed (see Appendix B for details of sea ice thickness specification).

2.  *Coupled ocean-atmosphere time slice.* What role does ocean-atmosphere coupling play in the response to sea ice? Previous studies have shown that such coupling potentially amplifies the response and produces additional impacts in remote regions including the tropics (Deser et al. 2015, 2016; Tomas et al. 2016; Smith et al. 2017; Oudar et al., 2017; Blackport and Russell, 2017). Coupled model simulations are therefore needed to assess the full response to sea ice. These experiments impose the same SIC fields as used in the atmosphere-only experiments (1.1, 1.5 to 1.8, see Appendix B for further details) allowing an assessment of the role of coupling. However, it is important to note that the background states are likely to be different between the coupled model and atmosphere-only simulations, and experiment set 4 is needed to isolate the effects of coupling (Smith et al. 2017).

3.  *Atmosphere-only time slice experiments to investigate regional forcing.* How and why does the atmospheric response to Arctic sea ice depend on the regional pattern of sea ice forcing? Previous studies have found that the atmospheric response is potentially very sensitive to the pattern of sea ice forcing (Sun et al 2015; Screen 2017). This sensitivity will be investigated by specifying SIC changes in two different regions, the Barents/Kara Seas and the Sea of Okhotsk. These regions represent the Atlantic and Pacific sectors which potentially produce opposite responses in the stratosphere (Sun et al. 2015), and have been highlighted as important regions by several studies (e.g. Honda et al. 1996; Petoukhov and Semenov 2010; Kim et al. 2014; Mori et al. 2014; Kug et al. 2015; Screen 2013, 2017).

4.  *Atmosphere-only time slice experiments to investigate the role of the model background state.* How and why does the atmospheric response to Arctic sea ice depend on the model background state? The atmospheric response to sea ice is potentially sensitive to the model background state (Balmaseda et al. 2010; Smith et al. 2017). This is investigated in experiment set 4 by repeating the atmosphere-only experiments 1.1 and 1.6 but specifying the climatological average SST obtained from the coupled model experiment (2.1) for the same model (as detailed in Appendix B), thereby imposing the coupled model biases. Analysis of the physical processes giving rise to sensitivity to background state could lead to an "emergent constraint" to determine the real world situation (Smith et al. 2017) as discussed further below. Furthermore, experiment sets (1), (2) and (4) together enable the role of coupling to be isolated, assuming the influences of coupling and background state are linear.

5.  *Atmosphere-only transient experiments.* What are the relative roles of SST and SIC in observed polar amplification over the recent period since 1979? These experiments are atmosphere-only simulations of the period since 1979. The control is a CMIP6 DECK experiment (Eyring et al. 2016) driven by the observed time series of monthly mean SST





and SIC. Replacing the monthly-mean time series with the climatological averages for SST and SIC separately enables the impacts of transient SST and SIC to be diagnosed. Individual years of interest, and the transient response to sea ice loss, may also be investigated.

6. *Coupled ocean-atmosphere transient experiments.* How does the global climate response to sea ice evolve on decadal and longer timescales? Previous studies suggest that the response to Arctic sea ice could be modulated by decadal and longer timescales changes in the ocean (Tomas et al. 2016; Sévellec et al. 2017). This may alter the pattern of tropical SST response which in turn may modulate changes in the Atlantic and Pacific ITCZs found in some studies (Chiang and Bitz 2005; Smith et al. 2017). Although experiment set 2 uses coupled models, the simulations are too short to capture changes in ocean circulation which occur on decadal and longer timescales. Hence experiment set 6 will investigate the decadal to centennial timescale response to sea ice using coupled model simulations in which sea ice is constrained to desired values (see Appendix B for further details).

A key goal of PAMIP is to determine the real world situation. Although the experiments are coordinated we anticipate a large spread in the model simulations. However, if the physical processes responsible for differences between models can be understood, then the model spread can be exploited to obtain an estimate of the real world situation using the concept of "emergent constraints" (Hall and Qu 2006; Collins et al. 2012; Bracegirdle and Stephenson 2013). For example, results using a single model but with different background states suggest that differences in the simulated Atlantic jet response to Arctic sea ice loss might be explained by the climatological planetary wave refractive index, observations of which suggest a moderate weakening of the jet in reality (Figure 9, Smith et al. 2017). The multi-model PAMIP simulations will be used to test the robustness of this relationship, and to identify other constraints to identify the real world's response to sea ice and the factors contributing to polar amplification.

# 6 Data request

The final definitive data request for PAMIP is documented in the CMIP6 data request, available at https://www.earthsystemcog.org/projects/wip/CMIP6DataRequest. The requested diagnostics are the same for all PAMIP experiments and will enable climate change and variability to be characterised and the underlying physical processes to be investigated. The basic set is based on the Decadal Climate Prediction Project (see Appendix D in Boer et al. 2016), though we stress that the data request is not intended to exclude other variables and participants are encouraged to retain variables requested by other MIPs if possible. Many studies have suggested that wave activity plays a key role in the atmospheric response to sea ice. However, the physical mechanism is poorly understood, with some studies highlighting increased wave activity (Jaiser et al. 2013; Kim et al. 2014; Peings and Magnusdottir 2014; Feldstein and Lee 2014; García-Serrano et al. 2015; Sun et al. 2015; Nakamura et al. 2015; Overland et al. 2016) and others showing reduced wave activity (Seierstad and Bader 2009; Wu and Smith 2016; Smith et al. 2017) in response to reduced Arctic sea ice. Hence, important additions to the





basic DCPP data request are monthly mean wave activity fluxes on pressure levels (Table 3) for diagnosing atmospheric zonal momentum transport as documented in the Dynamics and Variability MIP (DynVarMIP, see Gerber and Manzini 2016 for details on how to compute these variables).

## 7 Interactions with DECK and other MIPs

The CMIP6 DECK experiments (Eyring et al. 2016) characterise the sensitivity to external forcings and the levels of internal variability in each model, and therefore provide valuable information for interpreting the PAMIP experiments. The DECK AMIP experiment forms the basis for PAMIP experiment set 5, and the coupled model historical simulation is required to provide starting conditions for experiment sets 2 and 6.

PAMIP will compliment other CMIP6 MIPs, several of which will also provide valuable information on the causes and consequences of polar amplification. In particular, the magnitudes of polar amplification simulated by different models in response to future and past radiative forcings will be assessed from Scenario MIP (O'Neill et al. 2016) and Paleoclimate MIP (Kageyama et al. 2016). The roles of external factors including solar variability, volcanic eruptions, ozone, anthropogenic aerosols and greenhouse gases in driving polar amplification will be studied using experiments proposed in Detection and
Attribution MIP (Gillett et al. 2016), Geoengineering MIP (Kravitz et al. 2015), Aerosols and Chemistry MIP (Collins et al. 2017) and Volcanic Forcings MIP (Zanchettin et al. 2016); and the roles of AMV and PDV will be studied using experiments proposed by the Decadal Climate Prediction Project (DCPP, Boer et al. 2016) and Global Monsoons MIP (Zhou et al. 2016).

 The impacts of polar amplification on polar ice sheets and carbon uptake will be investigated by the Ice Sheet MIP (Nowicki
et al. 2016) and the Coupled Climate–Carbon Cycle MIP (Jones et al. 2016a), respectively. Some information on the impacts of reduced sea ice on the atmospheric circulation can be obtained from experiments proposed by Cloud Feedback MIP (CFMIP, Webb et al. 2017) in which an atmosphere model is run twice, forced by the same SSTs but different sea ice concentration fields. However, the CFMIP experiments are not specifically designed to investigate the response to sea ice, and interpreting them is complicated by the fact that the forcing fields will be different for each model.

 Improved understanding of the causes and consequences of polar amplification gained through PAMIP and other MIPs will help to interpret decadal predictability diagnosed in DCPP, and will reduce the uncertainties in future projections of climate change in polar regions and associated climate impacts, thereby contributing to the Vulnerability, Impacts, Adaptation, and Climate Services (VIACS) Advisory Board (Ruane et al. 2016).



## 8 Summary

Polar amplification – the phenomenon that external radiative forcing produces a larger change in surface temperature at high latitudes than the global average – is robustly simulated by climate models in response to increasing greenhouse gases. Polar amplification is projected to occur at both poles, but to be delayed in the Antarctic relative to the Arctic due to strong heat

uptake in the Southern Ocean. Arctic amplification appears to be already underway, with recent Arctic warming trends approximately twice as large as the global average and reductions in summer sea ice extent of more than 10% per decade. However, recent temperature trends in the Antarctic are non-uniform with warming over the Antarctic Peninsula and cooling elsewhere, and sea ice extent has actually increased slightly over recent decades in contrast to most climate model simulations.

Understanding the causes of polar amplification, and the drivers of recent trends in both the Arctic and Antarctic, represents a key scientific challenge and is important for reducing the uncertainties in projections of future climate change. Several different feedback mechanisms, operating at both high and low latitudes, have been identified but their relative roles are uncertain. Recent trends have also been influenced by external factors other than greenhouse gases, including aerosols, ozone and solar variations, and by changes in atmosphere and ocean circulations. A key uncertainty is the relative role of local processes that

directly affect the surface energy budget and remote processes that affect the poleward atmospheric heat transport. This balance helps to highlight the main feedbacks and processes that drive polar amplification, and can easily be assessed in numerical model experiments by separately imposing changes in sea ice concentration and remote sea surface temperatures (SSTs). Such experiments have been performed in recent studies (Screen et al. 2012; Perlwitz et al. 2015), but additional models are needed to obtain robust results. PAMIP will therefore provide a robust multi-model assessment of the roles of local sea ice and remote

SSTs in driving polar amplification. The tier 1 experiments focus on the changes between pre-industrial and present day, while lower tier experiments enable recent decades and future conditions to be investigated.

There is mounting evidence that polar amplification will affect the atmosphere and ocean circulation, with potentially important climate impacts in both the mid-latitudes and the tropics. In particular, polar amplification will reduce the equator

to pole surface temperature gradient which might be expected to weaken mid-latitude westerly winds, promoting cold winters in parts of Europe, North America and Asia. Furthermore, enhanced warming of the Northern Hemisphere relative to the Southern Hemisphere might be expected to shift tropical rainfall northwards, and potentially influence tropical storm activity. However, despite many studies and intensive scientific debate there is a lack of consensus on the impacts of reduced sea ice cover in climate model simulations, and the physical mechanisms are not fully understood. There are many potential reasons

for disparity across models, including differences in the magnitude and pattern of sea ice changes considered, how the changes are imposed, the use of atmosphere-only or coupled models, and whether other forcings such as greenhouse gases are included. Hence, coordinated model experiments are needed and will be performed in PAMIP. The tier 1 experiments involve atmosphere-only models forced with different combinations of sea ice and/or sea surface temperatures representing present





day, pre-industrial and future conditions. The use of three periods allows the signals of interest to be diagnosed in multiple ways. Lower tier experiments are proposed to investigate additional aspects and provide further understanding of the physical processes. Specific questions addressed by these are: What role does ocean-atmosphere coupling play in the response to sea ice? How and why does the atmospheric response to Arctic sea ice depend on the pattern of sea ice forcing? How and why

does the atmospheric response to Arctic sea ice depend on the model background state? What is the response to sea ice over the recent period since 1979? How does the response to sea ice evolve on decadal and longer timescales?

A key goal of PAMIP is to determine the real world situation. Although the experiments proposed here form a coordinated set, we anticipate large spread across models. However, this spread will be exploited by seeking "emergent constraints" in which

the real world situation is inferred from observations of a physical quantity that explains the model differences. For example, if differences in the mid-latitude wind response to Arctic sea ice are caused by differences in the refraction of atmospheric waves across models, then observations of the refractive index may provide a constraint on the real world response. In this way, improved process understanding gained through analysis of the unprecedented multi-model simulations generated by PAMIP will enable uncertainties in projections of future polar climate change and associated impacts to be reduced, and better

climate models to be developed.

## 9 Data availability

The model output from PAMIP will be distributed through the Earth System Grid Federation (ESGF) with digital object identifiers (DOIs) assigned. The list of requested variables, including frequencies and priorities, is given in Appendix B and has been submitted as part of the "CMIP6 Data Request Compilation". As in CMIP5, the model output will be freely accessible

through data portals after a simple registration process that is unique to all CMIP6 components. In order to document CMIP6's scientific impact and enable ongoing support of CMIP, users are requested to acknowledge CMIP6, the participating modelling groups, and the ESGF centres (see details on the CMIP Panel website at http://www.wcrp-climate. org/index.php/wgcm-cmip/about-cmip). Further information about the infrastructure supporting CMIP6, the metadata describing the model output, and the terms governing its use are provided by the WGCM Infrastructure Panel (WIP). Links to

this information may be found on the CMIP6 website. Along with the data itself, the provenance of the data will be recorded, and DOI's will be assigned to collections of output so that they can be appropriately cited. This information will be made readily available so that research results can be compared and the modelling groups providing the data can be credited.

The WIP is coordinating and encouraging the development of the infrastructure needed to archive and deliver the large amount

of information generated by CMIP6. In order to run the experiments, datasets for SST and SIC forcing, along with natural and anthropogenic forcings are required. SST and SIC forcing datasets for PAMIP are described in Appendix A and will be made available through the ESGF (https://esgf-node.llnl.gov/projects/input4mips/) with version control and DOIs assigned.





**Acknowledgements**.

Thanks to the US CLIVAR Working Group on Arctic Change and the Aspen Global Change Institute (AGCI) for hosting workshops which contributed to PAMIP planning, with funding from NASA, NSF, NOAA, and DOE. DMS was supported

by the joint DECC/Defra Met Office Hadley Centre Climate Programme (GA01101) and the EU H2020 APPLICATE project (GA727862). NCAR is sponsored by the National Science Foundation. JG-S was supported by the EU H2020 PRIMAVERA (GA641727) and Spanish MINECO-funded DANAE (CGL2015-68342-R) projects. JYoon was supported by the funding from the Korean Polar Research Institute through the grant PE16100. DM was supported by the EU H2020 Blue-Action (GA 727852) and BMBF project CLIMPRE InterDec (FKZ:01LP1609A). Thanks to Yannick Peings for creating the forcing fields.

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



**Table 1: PAMIP coordinated model experiments.** The contributions of local sea ice and remote SST to polar amplification, and the response to sea ice, will be diagnosed from atmosphere-only and coupled atmosphere ocean model experiments using different combinations of SST and SIC representing present day (pd), pre-industrial (pi) and future (fut, representing 2 degree warming) conditions. The signals of interest are obtained by differencing experiments, as shown in Table 2. Further details are given in appendix B.

| No. | Experiment name | Description | Notes | Tier | Start year | Number of years | Minimum ensemble size |
|---|---|---|---|---|---|---|---|
| **1. Atmosphere-only time slice experiments** | | | | | | | |
| 1.1 | pdSST-pdSIC | Time slice forced by climatological monthly mean SST and SIC for the present day (pd)[1,2] | Present day SST and SIC | 1 | 2000 | 1[2] | 100 |
| 1.2 | piSST-piSIC | Time slice forced by climatological monthly mean SST and SIC for pre-industrial (pi) conditions[3] | Pre-industrial SST and SIC | 2 | 2000 | 1 | 100 |
| 1.3 | piSST-pdSIC | Time slice forced by pi SST and pd SIC[3] | Different SST relative to 1.1 to investigate the role of SSTs in polar amplification | 1 | 2000 | 1 | 100 |
| 1.4 | futSST-pdSIC | Time slice forced by pd SIC and future SST representing 2 degree global warming (fut)[3] | | 2 | 2000 | 1 | 100 |
| 1.5 | pdSST-piArcSIC | Time slice forced by pd SST and pi Arctic SIC[3] | Different Arctic SIC relative to 1.1 to investigate the impacts of present day and future Arctic sea ice, and the role of Arctic SIC in polar amplification | 1 | 2000 | 1 | 100 |
| 1.6 | pdSST-futArcSIC | Time slice forced by pd SST and fut Arctic SIC[3] | | 1 | 2000 | 1 | 100 |
| 1.7 | pdSST-piAntSIC | Time slice forced by pd SST and pi Antarctic SIC[3] | Different Antarctic SIC relative to 1.1 to investigate the impacts of present day and future Antarctic sea ice, and the role of Antarctic SIC in polar amplification | 1 | 2000 | 1 | 100 |
| 1.8 | pdSST-futAntSIC | Time slice forced by pd SST and fut Antarctic SIC[3] | | 1 | 2000 | 1 | 100 |
| 1.9 | pdSST-pdSICSIT | Time slice forced by pd sea ice thickness (SIT) in addition to SIC and SST | Investigate the impacts of sea ice thickness changes | 3 | 2000 | 1 | 100 |
| 1.10 | pdSST-futArcSICSIT | Time slice forced by pd SST and fut Arctic SIC and SIT | Investigate the impacts of sea ice thickness changes | 3 | 2000 | 1 | 100 |
| **2. Coupled ocean-atmosphere time slice experiments** | | | | | | | |
| 2.1 | pa-pdSIC | Coupled time slice constrained by pd SIC[2,4,5] | | 2 | 2000 | 1 | 100 |
| 2.2 | pa-piArcSIC | Coupled time slice with pi Arctic SIC[3] | As 1.5 and 1.6 but with coupled model | 2 | 2000 | 1 | 100 |
| 2.3 | pa-futArcSIC | Coupled time slice with fut Arctic SIC[3] | | 2 | 2000 | 1 | 100 |
| 2.4 | pa-piAntSIC | Coupled time slice with pi Antarctic SIC[3] | As 1.7 and 1.8 but with coupled model | 2 | 2000 | 1 | 100 |
| 2.5 | pa-futAntSIC | Coupled time slice with fut Antarctic SIC[3] | | 2 | 2000 | 1 | 100 |
| **3. Atmosphere-only time slice experiments to investigate regional forcing** | | | | | | | |




| | | | | | | | |
|---|---|---|---|---|---|---|---|
| 3.1 | pdSST-futOkhotskSIC | Time slice forced by pd SST and fut Arctic SIC only in the Sea of Okhotsk | Investigate how the atmospheric response depends on the pattern of Arctic sea ice forcing | 3 | 2000 | 1 | 100 |
| 3.2 | pdSST-futBKSeasSIC | Time slice forced by pd SST and fut Arctic SIC only in the Barents/Kara Seas | | 3 | 2000 | 1 | 100 |
| **4. Atmosphere-only time slice experiments to investigate the role of the background state** | | | | | | | |
| 4.1 | modelSST-pdSIC | Time slice forced by pd SIC and pd SST from coupled model (2.1) rather than observations | In conjunction with experiments 1 and 2, isolate the effects of the background state from the effects of coupling | 3 | 2000 | 1 | 100 |
| 4.2 | modelSST-futArcSIC | Time slice forced by fut Arctic SIC and pd SST from coupled model (2.1) rather than observations | | 3 | 2000 | 1 | 100 |
| **5. Atmosphere-only transient experiments** | | | | | | | |
| 5.1 | amip-climSST | Repeat CMIP6 AMIP (1979-2014) but with climatological monthly mean SST | Use CMIP6 AMIP as the control. Investigate transient response, individual years, and the contributions of SST and SIC to recent climate changes | 3 | 1979 | 36 | 3 |
| 5.2 | amip-climSIC | Repeat CMIP6 AMIP (1979-2014) but with climatological monthly mean SIC | | 3 | 1979 | 36 | 3 |
| **6. Coupled ocean-atmosphere transient experiments** | | | | | | | |
| 6.1 | pa-pdSIC-ext | Coupled model extended simulation constrained with pd sea ice[4,6] | Experiments to investigate the decadal and longer impacts of Arctic and Antarctic sea ice. | 3 | 2000 | 100 | 1 |
| 6.2 | pa-futArcSIC-ext | Coupled model extended simulation constrained with fut Arctic sea ice[4,6] | | 3 | 2000 | 100 | 1 |
| 6.3 | pa-futAntSIC-ext | Coupled model extended simulation constrained with fut Antarctic sea ice[4,6] | | 3 | 2000 | 100 | 1 |

**Notes:**

Radiative forcing to be set to present day (year 2000) levels for all experiments except AMIP (5.1 and 5.2) where the CMIP6 protocol should be used.

5   [1] All necessary SST and sea ice fields will be provided to participants (Appendix A).

[2] Time slice simulations to begin on 1st April and run for 14 months. One year long runs are required to isolate short-term atmospheric responses from longer timescale ocean responses, which will be investigated separately (experiments 6).

[3] Past and future SIC and SST will be computed from the ensemble of CMIP5 projections (Appendix A). Sea ice thickness should be specified according to the CMIP6 AMIP protocol (Appendix B).

10   [4] Initial conditions for coupled model experiments to be taken from year 2000 of coupled model historical simulations.

[5] Sea ice concentration to be nudged into coupled model with a relaxation time-scale of 1 day

[6] Present day and future sea ice to be the same as used in experiments 1.1, 1.6 and 1.8. It is recommended to constrain sea ice by nudging but with a weak relaxation timescale of 2 months. However, appropriately calibrated long-wave fluxes applied to the sea ice model (following Deser et al. 2015) may also be used. See Appendix B for further discussion.





**Table 2: Suggested experiment combinations for analysis of PAMIP simulations.** The role of SST in polar amplification can be obtained by differencing experiments that have the same SIC but different SST (e.g. 1.1 and 1.3). The role of SIC in polar amplification, and the response to SIC, can be obtained by differencing experiments that have the same SST but different SIC (e.g. 1.1 and 1.5). Other combinations are shown in the table.

| | Polar amplification | | | | | | Response to sea ice | | |
| | Role of SST | | | Role of sea ice | | | Past | Future | Total |
| | Past | Future | Total | Past | Future | Total | | | |
|---|---|---|---|---|---|---|---|---|---|
| **Arctic** | **[1.1-1.3]**, [**1.5**-1.2], [AMIP-5.2] | [1.4-**1.1**] | [1.4-**1.3**] | **[1.1-1.5]**, [**1.3**-1.2], [AMIP-5.1] | **[1.6-1.1]**, [1.9-**1.1**] | **[1.6-1.5]**, [1.9-**1.5**] | **[1.1-1.5]**, [**1.3**-1.2], [2.1-2.2], [5.1-AMIP] | **[1.6-1.1]**, [2.3-2.1], [3.1-**1.1**], [3.2-**1.1**], [4.2-4.1], [6.2-6.1], [1.9-**1.1**] | **[1.6-1.5]**, [1.9-**1.5**], [2.3-2.2] |
| **Antarctic** | **[1.1-1.3]**, [**1.7**-1.2], [AMIP-5.2] | [1.4-**1.1**] | [1.4-**1.3**] | **[1.1-1.7]**, [**1.3**-1.2], [AMIP-5.1] | **[1.8-1.1]** | **[1.8-1.7]** | **[1.1-1.7]**, [**1.3**-1.2], [2.4-2.1], [5.1-AMIP] | **[1.8-1.1]**, [2.5-2.1], [6.3-6.1] | **[1.8-1.7]**, [2.5-2.4] |

**Notes:**
Experiments are defined in Table 1. AMIP is part of the CMIP6 DECK simulations (Eyring et al. 2016).
Tier 1 experiments are in **bold.**
Coupled model experiments are underlined.

**Table 3: Requested variables for diagnosing atmospheric zonal momentum transport.** Zonal mean variables (2-D, grid: YZT) on the plev39 grid (or a subset of plev39 for models which do contain all of the requested levels).

| Name | Long name [unit] |
|---|---|
| ua | eastward wind [m s$^{-1}$] |
| epfy | northward component of the Eliassen–Palm flux [m$^3$ s$^{-2}$] |
| epfz | upward component of the Eliassen–Palm flux [m$^3$ s$^{-2}$] |
| vtem | Transformed Eulerian mean northward wind [m s$^{-1}$] |
| wtem | Transformed Eulerian mean upward wind [m s$^{-1}$] |
| utendepfd | tendency of eastward wind due to Eliassen–Palm flux divergence [m s$^{-2}$] |





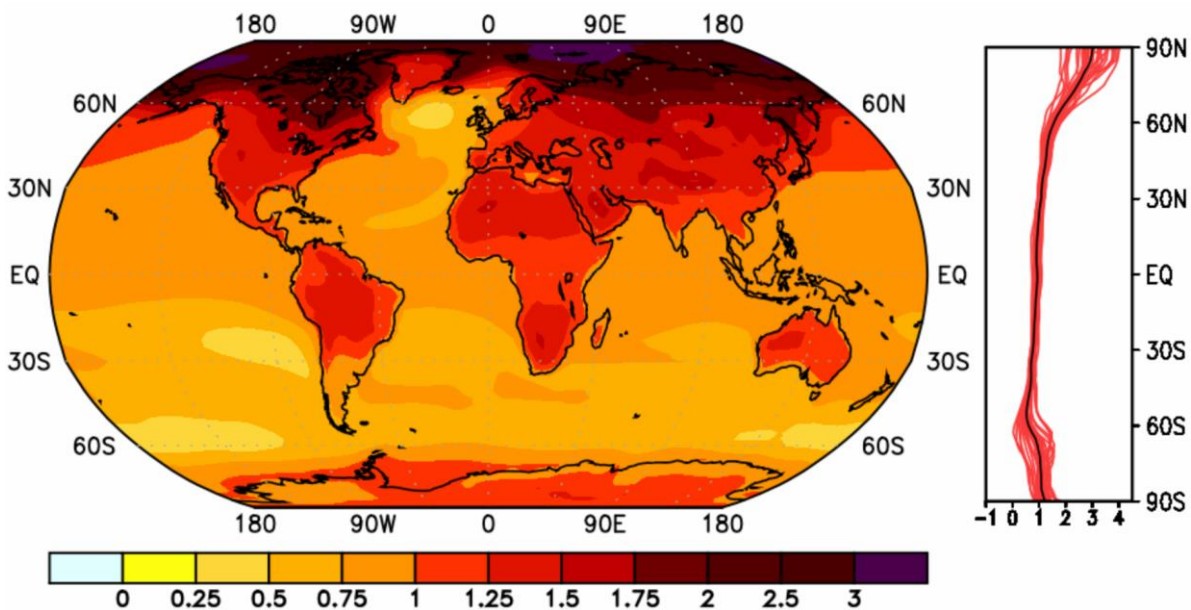

**Figure 1: Polar amplification in projections of future climate change.** Temperature change patterns derived from 31 CMIP5 model projections driven by RCP8.5, scaled to 1°C of global mean surface temperature change. The patterns have been calculated by computing 20-year averages at the end of the 21st (2080-2099) and 20th (1981-2000) centuries, taking their difference and normalizing it, grid-point by grid-point, by the corresponding value of global average temperature change for each model. The normalized patterns have then been averaged across models. The colour scale represents degrees Celsius per 1°C of global average temperature change. Zonal means of the geographical patterns are shown for each individual model (red) and for the multi-model ensemble mean (black).





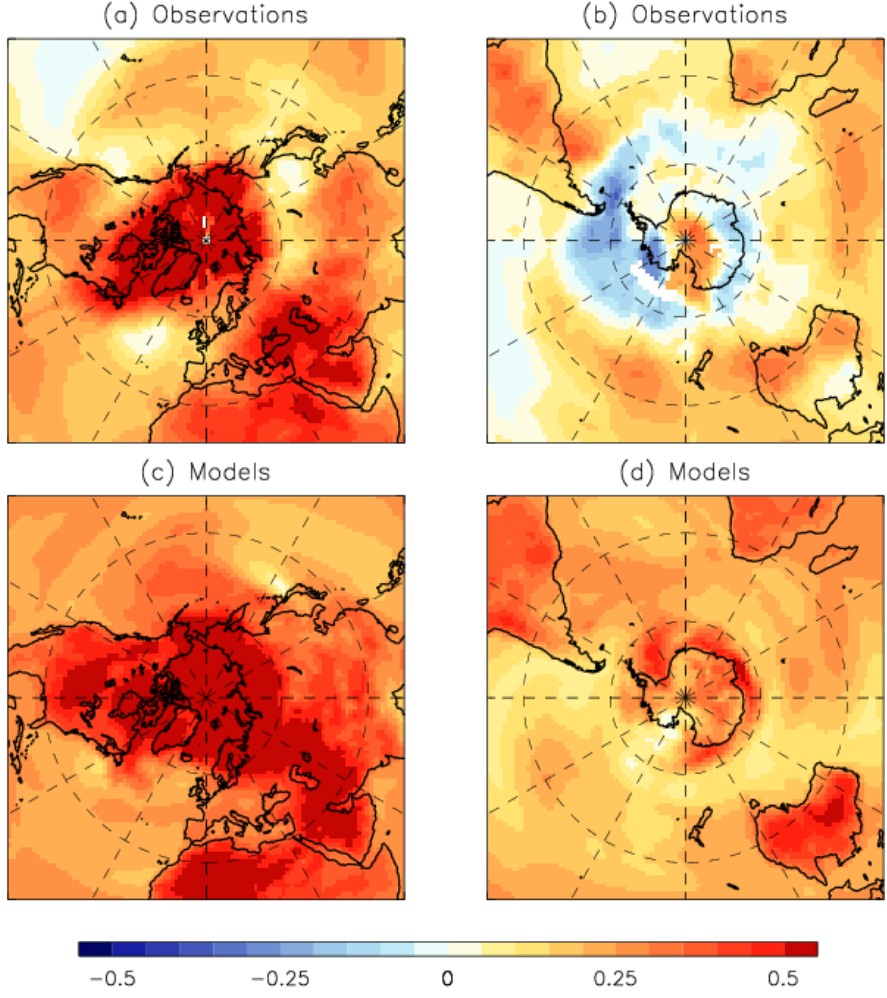

**Figure 2: Recent Arctic and Antarctic temperature trends (ºC per decade) in (a, b) observations and (c, d) model simulations.** Linear trends are shown for the 30 year period 1988 to 2017. Observations are taken as the average of HadCRUT4 (Morice et al. 2012), NASA-GISS (Hansen et al. 2010) and NCDC (Karl et al. 2015). Model trends are computed as the average from 25 CMIP5 model simulations driven by historical and RCP4.5 radiative forcings.





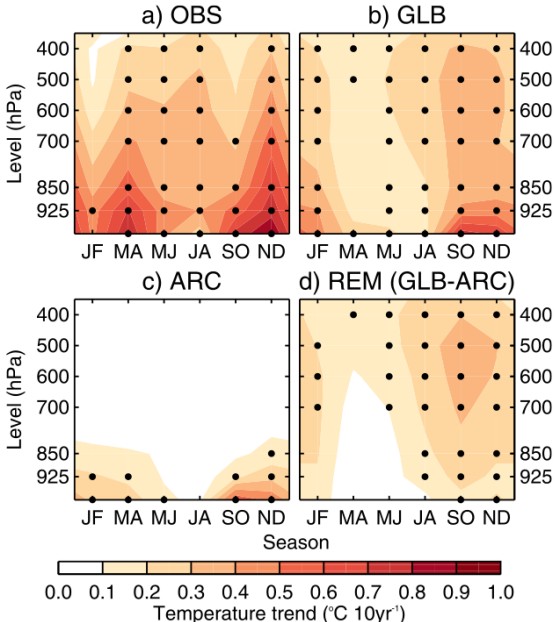

**Figure 3: Local and remote drivers of Arctic warming.** (a) Vertical and seasonal structure of the reanalysis ensemble-mean (OBS) Arctic-mean temperature trends (1979–2008). (b–d) As in (a), but for the model ensemble-mean simulations forced by observed changes in global SST and SIC (GLB), observed changes only in Arctic SIC and associated SST (ARC), and their difference (REM), respectively. Black dots show trends that are statistically significant at the 95% level (p < 0.05). These experiments enable the relative contributions of local (ARC) and remote (REM) processes to Arctic trends to be assessed, giving insight into the driving mechanisms. Source: Screen et al. 2012.

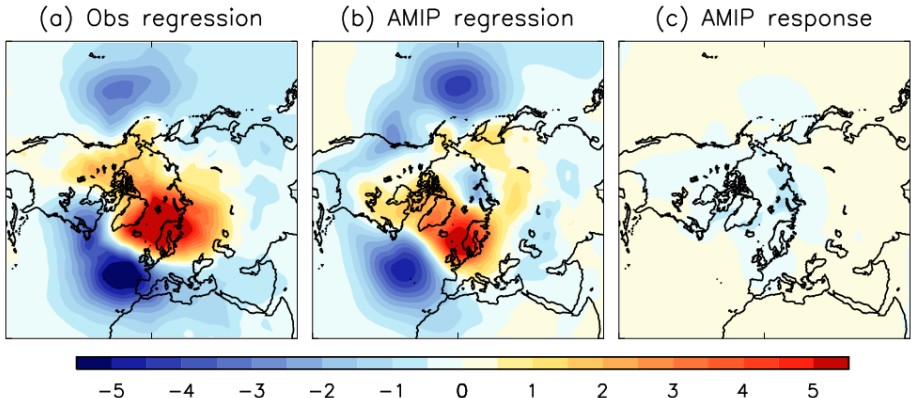

**Figure 4: Inability to diagnose atmospheric response to sea ice from observations alone.** Linear regression between autumn (September-November) Arctic sea ice extent and winter (December-February) mean sea level pressure (reversed sign) in (a) observations and (b) atmosphere model experiments forced by observed SIC and SST following the Atmosphere Model Intercomparison Project (AMIP) protocol. All time series were linearly detrended and cover the period December 1979 to November 2009. (c) Winter mean sea level response to reduced sea ice in atmospheric model experiments (scaled by the average autumn sea ice extent reduction). Units are hPa per million km$^2$. Source: Smith et al. 2017.



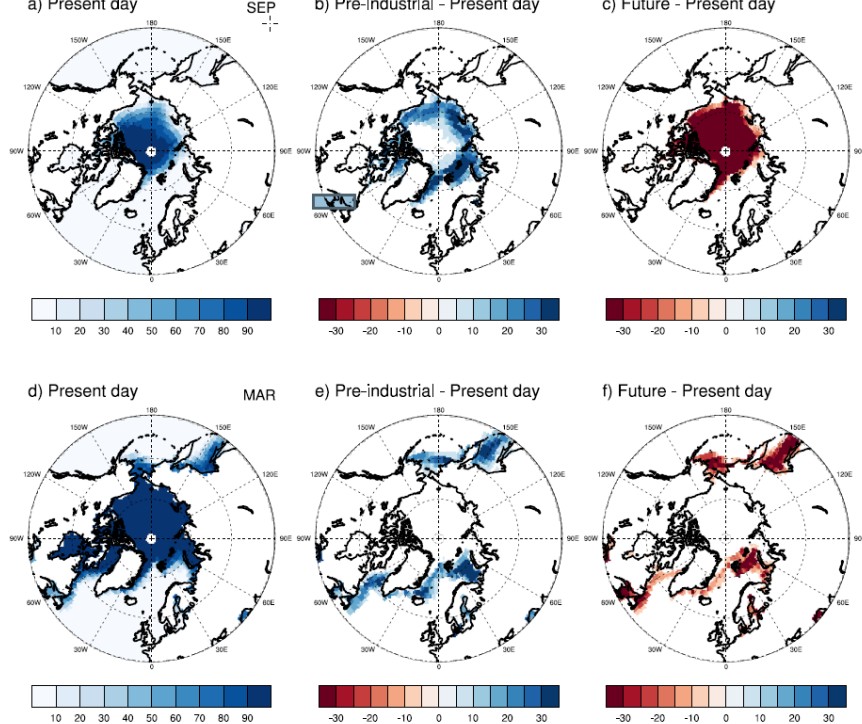

**Figure 5: Arctic sea ice forcing fields.** Present day Arctic sea ice concentration for (a) September and (d) March. Differences from present day fields are shown for (b, e) pre-industrial and (c, f) future conditions.





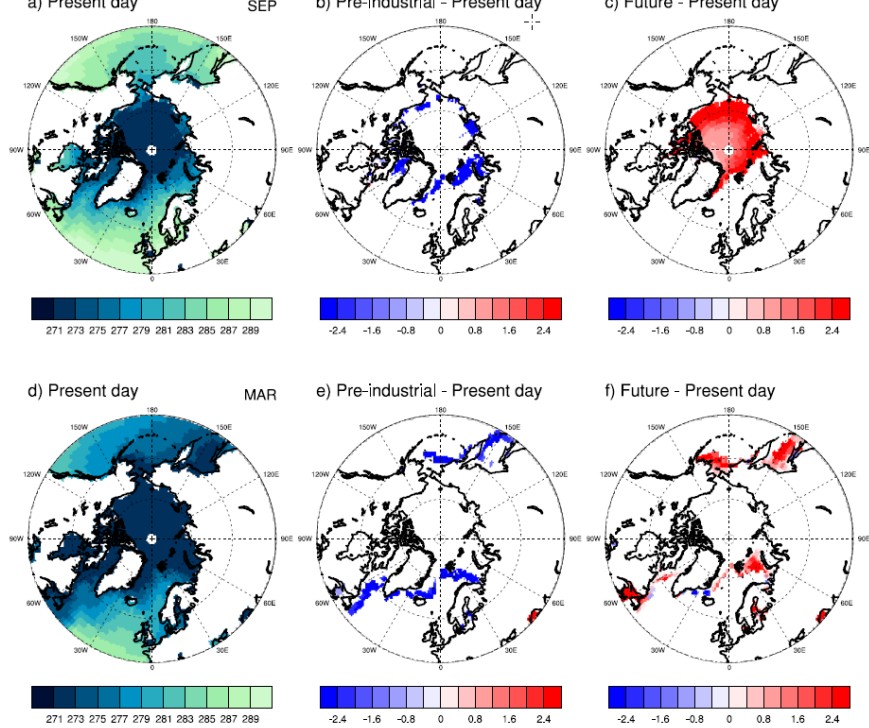

**Figure 6: Arctic SST forcing fields.** Present day Arctic SST for (a) September and (d) March. Differences from present day fields are shown for (b, e) pre-industrial and (c, f) future conditions




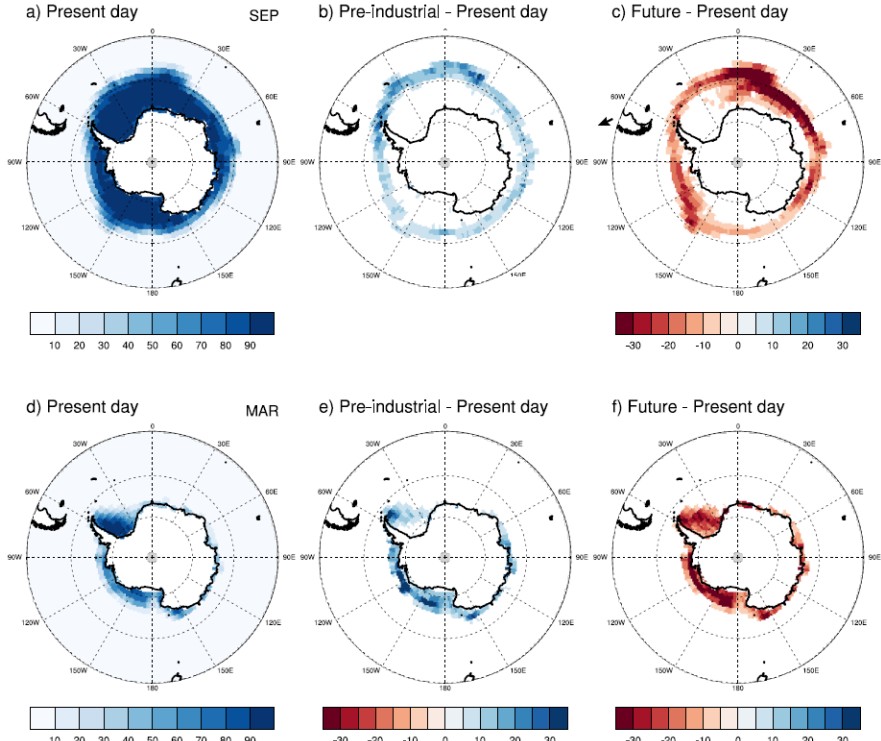

**Figure 7: Antarctic sea ice forcing fields.** Present day Antarctic sea ice concentration for (a) September and (d) March. Differences from present day fields are shown for (b, e) pre-industrial and (c, f) future conditions.



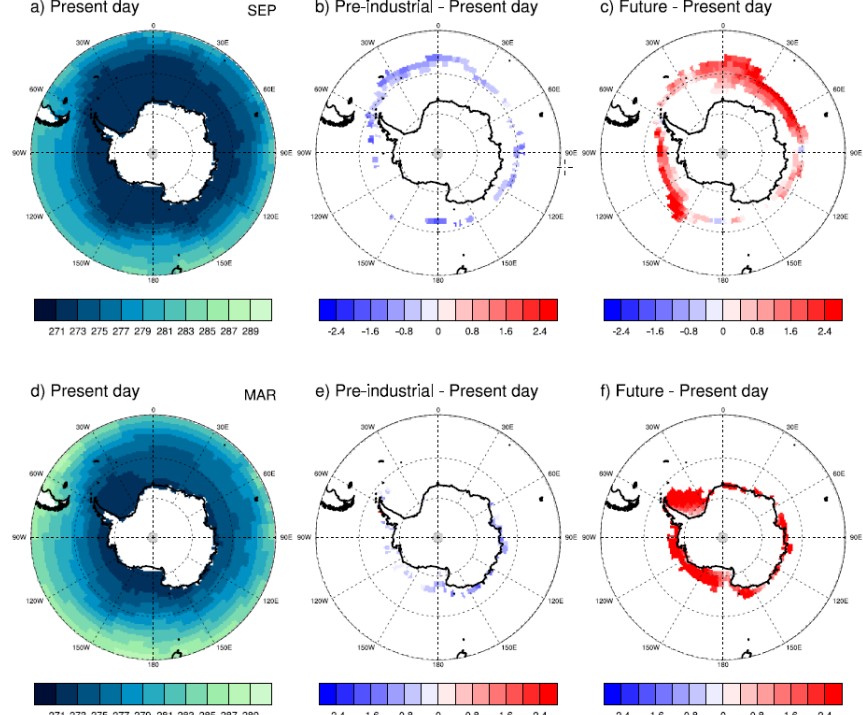

**Figure 8: Antarctic SST forcing fields.** Present day Antarctic SST for (a) September and (d) March. Differences from present day fields are shown for (b, e) pre-industrial and (c, f) future conditions.





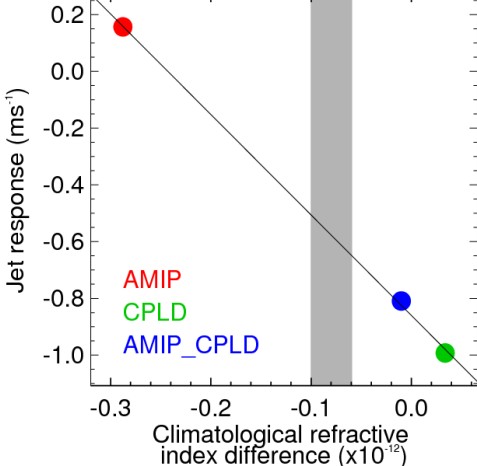

**Figure 9: Potential emergent constraint on atmospheric response to Arctic sea ice.** Dependence of Atlantic jet response to reduced Arctic sea ice on the background climatological refractive index difference between mid (25-35ºN) and high (60-80ºN) latitudes at 200 hPa. Grey shading shows the observed range from the ERA-Interim and NCEP II reanalyses. The Atlantic jet response is defined as the difference in zonal mean zonal wind at 200 hPa over the region 60-0ºW, 50-60ºN between model experiments with reduced and climatological Arctic sea ice. Experiments were performed with the same model but with three different configurations: atmosphere only (AMIP); fully coupled (CPLD); and atmosphere only but with SST biases from the coupled model (AMIP_CPLD). An "emergent constraint" is obtained where the observed refractive index difference (grey shading) intersects the simulated response (black line), suggesting a modest weakening of the Atlantic jet in response to reduced Arctic sea ice. Source: Smith et al. 2017.



**Appendix A: SIC, SIT and SST forcing data**

Forcing fields for the PAMIP experiments are available from the input4MIPs data server (https://esgf-node.llnl.gov/search/input4mips/). Filenames of forcing data for each PAMIP experiment are provided in Table A1. The derivation of forcing data is described here.

Monthly mean fields of SIC, SIT and SST are required for the present day, pre-industrial and future periods. For SST and SIC present day fields are obtained from the observations, using the 1979-2008 climatology from the Hadley Centre Ice and Sea Surface Temperature data set (HadISST, Rayner et al. 2003). For SIT over the Arctic, present-day fields are obtained from the Pan-Arctic Ice Ocean Modeling and Assimilation System (PIOMAS, Zhang and Rothrock, 2003). Future and pre-industrial

fields are obtained from the ensemble of 31 historical and RCP8.5 simulations from CMIP5 (see list of models in Table A2). However, models show a large spread in simulations of sea ice, so that a simple ensemble mean would produce an unrealistically diffuse ice edge. To obtain a more realistic ice edge we use present day observations to constraint the models, as follows:

- We define absolute global mean temperatures representing pre-industrial (13.67°C), present day (14.24°C) and future (2 degree warming, 15.67°C) periods. The present-day global mean temperature corresponds to the 1979-2008 average from HadCRUT4 observations (Morice et al. 2012). The pre-industrial global mean temperature is obtained by removing from this present day value an estimate of the global warming index (Haustein et al. 2017) for the period 1979-2008 (0.57°C). The future global mean temperature is defined as +2°C from the pre-industrial period, i.e.

15.67°C.

- For each model, find the periods when the 30 year mean global temperature equals the above values and compute the 30-year averaged fields.

- At each grid point, use the observed present day value to constrain the model simulations of future and pre-industrial conditions. This is achieved by computing a linear regression across the models between future (or pre-industrial)

and present day values simulated by the model ensemble, and taking the required future (or pre-industrial) estimate as the point where this regression relationship intersects the observed present day value. We use quantile regression rather than least squares regression to reduce the impact of outliers and hence provide a sharper ice edge. This procedure is used to create past and future SIC/SST/SIT fields, with the quantile of the regression being chosen to increase the signal. For the future, the upper (lower) quantile regression is used for SIC/SIT (SST), in order to give

more weight to models with less sea ice and warmer SST. Conversely, for the pre-industrial period, the lower (upper) quantile is used for SIC/SIT (SST), giving more weight to models with larger sea ice and cooler SST.



Some experiments, such as 1.6, require SSTs to be specified in regions where the sea ice has been removed. We follow the methodology of Screen et al. (2013), i.e. we impose pre-industrial/future SST (derived from the quantile regression) in grid points where pre-industrial/future SIC deviate by more than 10% of its present-day value. Example SIC and SST fields are shown in Figures 5 to 8.

In experiments 3, future sea ice changes are only imposed in the Barents/Kara Seas and Sea of Okhotsk. Future SIC fields for these experiments were created by replacing present day values with future values but only in the required regions, defined as [10ºE-110ºE, 65ºN-85ºN] for the Barents/Kara Seas and [135ºE-165ºE, 40ºN-63ºN] for the Sea of Okhotsk.

10   For experiments 5 monthly mean SST and SIC climatologies were created from the CMIP6 AMIP data. For 5.1, SST is set to the transient values where sea ice substantially deviates (by more than 10%) from climatology. For 5.2, SST is set to the transient values where sea ice is absent in the climatology but present in the transient fields, and -1.8ºC where sea ice is present in the climatology but absent in the transient fields.

15   For experiments 1.9 and 1.10, SIT in the Arctic is derived from PIOMAS. Since no SIT observations are available in the Antarctic, we use the median of present day values simulated by the model ensemble. The same present-day SIT values are used in the Antarctic in the SIT forcing field of experiments 6.1 and 6.2. For experiment 6.3 (future Antarctic SIC/SIT), we use the lower quartile of future values simulated by the model ensemble. Where SIC is greater than 15%, but SIT equals 0, SIT is set to 15cm.

**Table A1: Names of forcing files for each experiment.** Files are available from the input4MIPs data server (https://esgf-node.llnl.gov/search/input4mips/). Comments in square brackets are optional guidance for groups that are able to constrain sea ice thickness (sit).

| No. | Experiment name | Names of forcing files |
|---|---|---|
| 1.1 | pdSST-pdSIC | tos_input4MIPs_SSTsAndSeaIce_PAMIP_pdSST_pdSIC_gn.nc<br>sic_input4MIPs_SSTsAndSeaIce_PAMIP_pdSST_pdSIC_gn.nc |
| 1.2 | piSST-piSIC | tos_input4MIPs_SSTsAndSeaIce_PAMIP_piSST_piSIC_gn.nc<br>sic_input4MIPs_SSTsAndSeaIce_PAMIP_piSST_piSIC_gn.nc |
| 1.3 | piSST-pdSIC | tos_input4MIPs_SSTsAndSeaIce_PAMIP_piSST_pdSIC_gn.nc<br>sic_input4MIPs_SSTsAndSeaIce_PAMIP_pdSST_pdSIC_gn.nc |
| 1.4 | futSST-pdSIC | tos_input4MIPs_SSTsAndSeaIce_PAMIP_fuSST_pdSIC_gn.nc<br>sic_input4MIPs_SSTsAndSeaIce_PAMIP_pdSST_pdSIC_gn.nc |
| 1.5 | pdSST-piArcSIC | tos_input4MIPs_SSTsAndSeaIce_PAMIP_pdSST_piSIC_Arctic_gn.nc<br>sic_input4MIPs_SSTsAndSeaIce_PAMIP_pdSST_piSIC_Arctic_gn.nc |
| 1.6 | pdSST-futArcSIC | tos_input4MIPs_SSTsAndSeaIce_PAMIP_pdSST_fuSIC_Arctic_gn.nc<br>sic_input4MIPs_SSTsAndSeaIce_PAMIP_pdSST_fuSIC_Arctic_gn.nc |
| 1.7 | pdSST-piAntSIC | tos_input4MIPs_SSTsAndSeaIce_PAMIP_pdSST_piSIC_Antarctic_gn.nc<br>sic_input4MIPs_SSTsAndSeaIce_PAMIP_pdSST_piSIC_Antarctic_gn.nc |



| 1.8 | pdSST-futAntSIC | tos_input4MIPs_SSTsAndSeaIce_PAMIP_pdSST_fuSIC_Antarctic_gn.nc |
| | | sic_input4MIPs_SSTsAndSeaIce_PAMIP_pdSST_fuSIC_Antarctic_gn.nc |
| 1.9 | pdSST-pdSICSIT | tos_input4MIPs_SSTsAndSeaIce_PAMIP_pdSST_pdSIC_gn.nc |
| | | sic_input4MIPs_SSTsAndSeaIce_PAMIP_pdSST_pdSIC_gn.nc |
| | | sit_input4MIPs_SSTsAndSeaIce_PAMIP_pdSST_pdSIC_pdSIT_gn.nc |
| 1.10 | pdSST-futArcSICSIT | tos_input4MIPs_SSTsAndSeaIce_PAMIP_pdSST_fuSIC_Arctic_gn.nc |
| | | sic_input4MIPs_SSTsAndSeaIce_PAMIP_pdSST_fuSIC_Arctic_gn.nc |
| | | sit_input4MIPs_SSTsAndSeaIce_PAMIP_pdSST_fuSIC_fuSIT_Arctic_gn.nc |
| 2.1 | pa-pdSIC | sic_input4MIPs_SSTsAndSeaIce_PAMIP_pdSST_pdSIC_gn.nc |
| | | [sit as used in experiment 1.1] |
| 2.2 | pa-piArcSIC | sic_input4MIPs_SSTsAndSeaIce_PAMIP_pdSST_piSIC_Arctic_gn.nc |
| | | [sit as used in experiment 1.5] |
| 2.3 | pa-futArcSIC | sic_input4MIPs_SSTsAndSeaIce_PAMIP_pdSST_fuSIC_Arctic_gn.nc |
| | | [sit as used in experiment 1.6] |
| 2.4 | pa-piAntSIC | sic_input4MIPs_SSTsAndSeaIce_PAMIP_pdSST_piSIC_Antarctic_gn.nc |
| | | [sit as used in experiment 1.7] |
| 2.5 | pa-futAntSIC | sic_input4MIPs_SSTsAndSeaIce_PAMIP_pdSST_fuSIC_Antarctic_gn.nc |
| | | [sit as used in experiment 1.8] |
| 3.1 | pdSST-futOkhotskSIC | tos_input4MIPs_SSTsAndSeaIce_PAMIP_pdSST_fuSIC_Okhotsk_gn.nc |
| | | sic_input4MIPs_SSTsAndSeaIce_PAMIP_pdSST_fuSIC_Okhotsk_gn.nc |
| 3.2 | pdSST-futBKSeasSIC | tos_input4MIPs_SSTsAndSeaIce_PAMIP_pdSST_fuSIC_BK_sea_gn.nc |
| | | sic_input4MIPs_SSTsAndSeaIce_PAMIP_pdSST_fuSIC_BK_sea_gn.nc |
| 4.1 | modelSST-pdSIC | tos to be created from experiment 2.1 as described in Appendix B |
| | | sic_input4MIPs_SSTsAndSeaIce_PAMIP_pdSST_pdSIC_gn.nc |
| 4.2 | modelSST-futArcSIC | tos to be created from experiment 2.1 as described in Appendix B |
| | | sic_input4MIPs_SSTsAndSeaIce_PAMIP_pdSST_fuSIC_Arctic_gn.nc |
| 5.1 | amip-climSST | tos_input4MIPs_SSTsAndSeaIce_PAMIP_pdSST_trSIC_gn.nc |
| | | sic_input4MIPs_SSTsAndSeaIce_PAMIP_pdSST_trSIC_gn.nc |
| 5.2 | amip-climSIC | tos_input4MIPs_SSTsAndSeaIce_PAMIP_trSST_pdSIC_gn.nc |
| | | sic_input4MIPs_SSTsAndSeaIce_PAMIP_trSST_pdSIC_gn.nc |
| 6.1 | pa-pdSIC-ext | sic_input4MIPs_SSTsAndSeaIce_PAMIP_pdSST_pdSIC_gn.nc |
| | | [sit_input4MIPs_SSTsAndSeaIce_PAMIP_pdSST_pdSIC_pdSIT_gn.nc] |
| 6.2 | pa-futArcSIC-ext | sic_input4MIPs_SSTsAndSeaIce_PAMIP_pdSST_fuSIC_Arctic_gn.nc |
| | | [sit_input4MIPs_SSTsAndSeaIce_PAMIP_pdSST_fuSIC_fuSIT_Arctic_gn.nc] |
| 6.3 | pa-futAntSIC-ext | sic_input4MIPs_SSTsAndSeaIce_PAMIP_pdSST_fuSIC_Antarctic_gn.nc |
| | | [sit_input4MIPs_SSTsAndSeaIce_PAMIP_pdSST_fuSIC_fuSIT_Antarctic_gn.nc] |

**Table A2: List of models used to construct the forcing fields**

| Acronym | Research Center |
|---|---|
| ACCESS1-0 | Commonwealth Scientific and Industrial Research Organisation, and Bureau of Meteorology, Australia |
| ACCESS1-3 | |
| bcc-csm1-1 | Beijing Climate Center, China |
| bcc-csm1-1-m | |
| CanESM2 | Canadian Center for Climate Modeling and Analysis, Canada |





| | |
|---|---|
| CCSM4 | National Center for Atmospheric Research, USA |
| CESM1-BGC | |
| CESM1-CAM5 | |
| CMCC-CM | Centro Euro-Mediterraneo per I Cambiamenti Climatici, Italy |
| CMCC-CMS | |
| CNRM-CM5 | Centre National de Recherches Meteorologiques / Centre Europeen de Recherche et Formation Avancees en Calcul Scientifique, France |
| CSIRO-Mk3-6-0 | Commonwealth Scientific and Industrial Research Organisation, in collaboration with the Queensland Climate Change Centre of Excellence, Australia |
| EC-EARTH | EC-Earth |
| FIO-ESM | The First Institute of Oceanography, China |
| GFDL-CM3 | U.S. Department of Commerce/National Oceanic and Atmospheric Administration/Geophysical Fluid Dynamics Laboratory, USA |
| GFDL-ESM2G | |
| GFDL-ESM2M | |
| GISS-E2-H | National Aeronautics and Space Administration/ Goddard Institute for Space Studies, USA |
| GISS-E2-R | |
| HadGEM2-CC | Met office Hadley Centre, UK |
| HadGEM2-ES | |
| inmcm4 | Institute for Numerical Mathematics, Russia |
| IPSL-CM5A-LR | Institut Pierre Simon Laplace, France |
| IPSL-CM5A-MR | |
| IPSL-CM5B-LR | |
| MIROC5 | Center for Climate System Research (University of Tokyo), National Institute for Environmental Studies,  and Frontier Research Center for Global Change, Japan |



| MPI-ESM-LR | Max Planck Institute for Meteorology, Germany |
|---|---|
| MPI-ESM-MR | |
| MRI-CGCM3 | Meteorological Research Institute, Japan |
| NorESM1-M | Norwegian Climate Centre, Norway |
| NorESM1-ME | |

## Appendix B: Experiment details

### AMIP II

Before use, the forcing data (SST, SIC and SIT) should be processed following the standard AMIP II protocol (Taylor et al.
2000). This is to ensure that monthly means computed from the model (after interpolating to the required model timesteps)
agree with the monthly means in the forcing files.

### Radiative forcing

Present day radiative forcings, taken as the year 2000, should be used for all experiments except set 5 for which time varying
forcings for the period 1979 to 2014 should be specified in accordance with the AMIP protocol (Eyring et al. 2016).

### Start date and length of simulations

Experiments 1 to 4 should start on 1$^{st}$ April 2000 and run for 14 months, with the first two months ignored to allow for an
initial model spin up. Experiments 6 start at the same time but extend to 100 years. Experiments 5 start on 1$^{st}$ January 1979
and end on 31$^{st}$ December 2014 in accordance with the AMIP protocol (Eyring et al. 2016).

### Initial conditions and ensemble generation

Initial conditions for atmosphere only experiments 1, 3 and 4 should be based on the AMIP simulation for 1$^{st}$ April 2000 if
possible, though any suitable start dump may be used, noting that the first two months of the simulations will be ignored.
Initial conditions for the coupled experiments 2 and 6 should be based on 1$^{st}$ April 2000 from the CMIP6 historical simulation
(Eyring et al. 2016). Ideally, different ocean states will be sampled by using different ensemble members of the historical
simulation if these are available. Large ensembles (~100 members) are requested in order to obtain statistically robust results
since models typically simulate a small atmospheric response to sea ice relative to internal variability (Screen et al. 2014; Mori
et al. 2014). We note that this may not be the case in reality, since models underestimate the signal to noise ratio in seasonal
and interannual forecasts of the NAO (Scaife et al. 2014; Eade et al. 2014; Dunstone et al. 2016). The results are not expected



to be particularly sensitive to the way in which ensemble members are generated, and any suitable method may be used but should be documented.

**Frequency of boundary conditions**

SST and sea ice boundary conditions are specified as monthly means, and should be interpolated to the required model timestep
(as is standard practice). Daily boundary conditions might strengthen some of the signals, but the additional complexity was considered unnecessary for assessing the physical processes and signals of interest here.

**Constraining sea ice in coupled models**

It is important that the sea ice fields used in the coupled model experiment set 2 are close to those used in the atmosphere-only simulations (1 and 4) so that differences are not caused by different sea ice forcing fields. It is therefore recommended that sea
ice concentrations are nudged into the coupled model, with a strong relaxation timescale (or equivalent restoring flux) of one day. However, decadal and longer timescale responses investigated in experiments 6 could potentially be contaminated by undesired responses to the nudging increments. It is therefore recommended that a weaker relaxation timescale of 2 months is used for these. This is similar to the DCPP component C experiments (Boer et al. 2016) which investigate the response to AMV and PDV, and technical issues relating to nudging are discussed in Technical Note 2 available from the DCPP website
(http://www.wcrp-climate.org/dcp-overview). Similar to the DCPP experiments, groups are recommended to monitor their experiments and take action, perhaps by reducing the relaxation timescale or applying balancing increments, to avoid unrealistic responses. Alternatively, appropriately calibrated long-wave fluxes applied to the sea ice model (following Deser et al. 2015) may also be used in experiments 6, but the calibration procedure should be documented. We note that all approaches for constraining sea ice are imperfect, but experiments 6 will nevertheless provide important information on the transient
response that is not available from the other experiments.

**Sea ice thickness**

Some participating groups may not able to specify sea ice thickness. Hence, in the atmosphere-only experiments (except 1.9 and 1.10) sea ice thickness should be treated in the same way as in the AMIP DECK simulation. We note that there is not a common protocol, but in practice the sea ice thickness will be at least 2m so that differences in surface fluxes between models
will be small. Experiments 1.9 and 1.10 are designed to investigate the impacts of future sea ice thickness changes, and sea ice thickness should be constrained with a relaxation timescale (or equivalent flux) of 5 days. Groups that are able to specify sea ice thickness are requested to do so for the coupled model experiments, using values from the equivalent atmosphere-only simulations for experiments 2, and the fields provided by PAMIP for experiments 6. If sea ice thickness cannot be specified then it should be left free to evolve in the coupled model experiments.



**SST forcing fields for experiment set 4**

Experiments 4.1 and 4.2 repeat experiments 1.1 and 1.6 but with present day SSTs taken from the coupled model experiment 2.1 instead of from observations. This allows the sensitivity to background SSTs to be investigated, and the role of coupling to be isolated (assuming signals add linearly). Experiments 4.1 and 4.2 use the same SIC forcing fields as experiments 1.1 and

5    1.6, but participants will need to create their own monthly mean present day SST forcing fields by taking the ensemble average for each month from experiment 2.1. Experiment 4.2 requires SSTs to be specified in regions where the sea ice has been removed. It is critically important that the change in SST in these regions between experiments 4.2 and 4.1 is exactly the same as between experiments 1.6 and 1.1, so that the forcings are identical. Hence, in regions where sea ice has been removed, SST in experiment 4.2 should be set equal to the SST in experiment 4.1 plus the difference in SST between experiments 1.6 and

10    1.1 (i.e. experiment 1.6 minus experiment 1.1).