# Peer review of "The Polar Amplification Model Intercomparison Project (PAMIP) contribution to CMIP6: investigating the causes and consequences of polar amplification"

_Geoscientific Model Development, 2018_

## Referee Comment (RC1) · Anonymous Referee #1 · 26 Jun 2018

This manuscript outlines the rationale and protocol for the Polar Amplification MIP, one of the many CMIP6 contributed MIPs. Overall, the paper is clearly written and provides sound rationale for the proposed experiments. I only have a few technical comments.

1. Page 3, Line 30: Huang, Xia and Tan (JGR, 2017) argue that the pattern of $CO_2$ radiative forcing also contributes to polar amplification.

2. Page 5, Line 19: England, Polvani and Sun (JCLIM 2018) have also shown an equatorward shift of the SH jet in response to Antarctic sea ice loss.

[Figure]

3. Page 8, Line 31: The proposed AMIP-style experiments (in experiment set 1) aim to investigate the relative contributions of local SIC changes and remote SST changes to polar amplification, but there is no experiment that quantifies the total polar amplification. Is it safe to assume that the SIC and SST perturbation experiments are linearly additive?

4. Page 9, Line 11: Should be Blackport and Kushner

5. Page 12, Line 21: Based on Table 1, seems like most of these are tier 1.

6. Page 13, Line 12: Observations are not a "control" experiment - how do we know that the climatological refractive index that one would compute from observations is not already perturbed by climate change?

7. Table 1: Experiment set 2 - what does 'pa' stand for?

8. Table 2: I do not understand why 1.9-1.1 is a in the future column. Also, there is no reference to experiment 1.10 anywhere in this table.

9. Figure 1: How do we define polar amplification? as a mean of ratios or a ratio of means? Hind et al., Scientific Reports (2016).

10. Figure 2: 31 models are used in Figure 1 but only 25 in Figure 2. Why?

11. Page 36, Lines 29-31: Maybe I am mixing things up, but, shouldn't this read this way: "For future, the **lower (upper)** quartile regression is used for SIC/SIT (SST), in order to give more weight to models with less sea ice and warmer SST. Conversely, for the pre-industrial period, the **upper (lower)** quartile is used ..."

---

## Referee Comment (RC2) · Anonymous Referee #2 · 13 Jul 2018

This manuscript proposed several experiments for the PAMIP, one of the subset of CMIP6, in order to better understand the causes and consequences of polar amplification. I find this paper is well-written and only have some minor comments, as follows:

1. Caveat on AMIP simulations (tier 1 and tier 5) One interesting but controversial question in this area is whether observed recent Arctic sea ice loss since 1979 has significantly affected the jet stream and caused more winter cold extremes in the mid-latitudes (or the debate on "Warm Arctic, Cold Continents"). I am just wondering if PAMIP wants to touch on this question in tier 1. Tier 5 is very nice, but I am a little

worried that the member size might be not enough to derive the forced sea ice loss effect for individual models. Sun et al. (2016) analyzed 50-member AMIP experiments to examine the "Warm Arctic, Cold Continents" hypothesis. Initially when they only had 10-member simulations, there was some cooling over Eurasia in response to Arctic sea ice loss. However, it turns out that is just internal variability because the cooling signal disappears when the ensemble size increases to 30. Similarly here, I suspect that 3 members might be not enough to show the forced response to Arctic sea ice loss for a single model. In other words, one should be able to use all available model ensemble to examine overall whether models show any response signal to observed sea ice loss but we may not be able to look at the sea ice loss effect in single models because of the large internal variability.

2. Caveat on the coupled model experiments (tier 2 and tier 6). The design of tier 2 experiment appears to be different from most previous fully coupled ocean-atmosphere studies (Deser et al. 2015, 2016; Tomas et al. 2016; Smith et al. 2017; Oudar et al., 2017; Blackport and Kushner, 2017). From the point view of attribution study, I think the tier 6 more represents the full sea ice loss effect because in reality and in projected CMIP simulations the ocean will be allowed to justify when the sea ice declines (in other words sea ice reduction occurs gradually and ocean can adjust to that). Given the reason above, I feel that tier 2 can be used to demonstrate the role of quick ocean-atmosphere coupling but one has to look at tier 6 to understand the role of full ocean-atmosphere coupling (tier 2 might be more like slab-ocean coupled results to some extent).

Another small point: 100 years appear to be too short for tier 6. 200 years might be necessary because the first 100 years or so might need to be discarded due to model spin-up (ocean adjustment).

3. Page 5 lines 5, 10: Liu et al. 2012 and Mori et al. 2014 are both modeling studies. Therefore, they probably should go to the next lines on "diverge considerable on the dynamical response", together with other studies that suggested recent Arctic sea ice

loss did not significantly affect the jet stream and "Warm Arctic, Cold Continents" may simply reflect the atmospheric internal variability (e.g. Sun et al. 2016; Ogawa et al. 2018). These are comparable studies which all conducted AMIP-type simulations by prescribing recent Arctic sea ice loss.

4. Page 7 line 10: isolating the response to sea ice (loss or change?). Also maybe I didn't fully understand the argument of "additional forcings". I thought we are discussing the effect of sea ice loss-alone. McCusker et al. (2017) have presented evidence that the atmospheric circulation response to sea ice loss and greenhouse gas forcing are remarkably linear. Thus it probably does not matter much.

5. Page 9 Line 10: Blackport and Kushner, 2017

6. Figure 5b): there is a weird box in the bottom left

References:

McCusker, K.E., Kushner, J.C. Fyfe, M. Sigmond, V.V. Kharin and C.M. Bitz, (2017): Remarkable separability of circulation response to Arctic sea ice loss and greenhouse gas forcing, Geophys. Res. Lett., 44, 7955-7964, doi: 10.1002/2017gl074327.

Ogawa, F., Keenlyside, N., Gao, Y., Koenigk, T., Yang, S., Suo, L., et al. (2018). Evaluating impacts of recent Arctic sea ice loss on the northern hemisphere winter climate change. Geophysical Research Letters, 45, 3255–3263. https://doi.org/10.1002/2017GL076502

Sun, L., J. Perlwitz and M. Hoerling (2016): What caused the recent "Warm Arctic, Cold Continents" trend pattern in winter temperatures?, Geophys. Res. Lett., 43, doi:10.1002/2016GL069024.

---

## Author Comment (AC1) · 29 Aug 2018

Many thanks for your comments. Please see our replies below.

Anonymous Referee #1 This manuscript outlines the rationale and protocol for the Polar Amplification MIP, one of the many CMIP6 contributed MIPs. Overall, the paper is clearly written and provides sound rationale for the proposed experiments. I only have a few technical comments.

1. Page 3, Line 30: Huang, Xia and Tan (JGR, 2017) argue that the pattern of $CO_2$

radiative forcing also contributes to polar amplification.

Thanks – we have now noted this in the manuscript.

2. Page 5, Line 19: England, Polvani and Sun (JCLIM 2018) have also shown an equatorward shift of the SH jet in response to Antarctic sea ice loss.

Reference now included – thanks.

3. Page 8, Line 31: The proposed AMIP-style experiments (in experiment set 1) aim to investigate the relative contributions of local SIC changes and remote SST changes to polar amplification, but there is no experiment that quantifies the total polar amplification. Is it safe to assume that the SIC and SST perturbation experiments are linearly additive?

The total impact of SIC and SST changes can be diagnosed by comparing experiments 1.1 and 1.2. This can be used to test the linearity of SIC and SST perturbations, at least for differences between pre-industrial and present day conditions. We have now noted this in the text.

4. Page 9, Line 11: Should be Blackport and Kushner

Corrected – thanks.

5. Page 12, Line 21: Based on Table 1, seems like most of these are tier 1.

We are not sure what you mean here. Assessing the role of SST in future polar amplification requires tier 2 experiment 1.4, and assessing recent decades requires the tier 3 atmosphere-only transient experiments (5.1 and 5.2). Note that the experiment set (1 to 6) is not the same as the tier, which is listed in column 5 of Table 1.

6. Page 13, Line 12: Observations are not a "control" experiment - how do we know that the climatological refractive index that one would compute from observations is not already perturbed by climate change?

We assume that the model simulations of the present day will include the effects of climate change on the refractive index and so may be compared with observations (to produce the x-axis in Fig. 9).

7. Table 1: Experiment set 2 - what does 'pa' stand for?

The prefix "pa" was suggested by the CMIP6 data panel to denote partially coupled experiments that are unique to PAMIP. We have clarified this in the table caption.

8. Table 2: I do not understand why 1.9-1.1 is a in the future column. Also, there is no reference to experiment 1.10 anywhere in this table.

This was a mistake and has now been corrected to read 1.10-1.9. Many thanks for drawing our attention to this.

9. Figure 1: How do we define polar amplification? as a mean of ratios or a ratio of means? Hind et al., Scientific Reports (2016).

How to quantify polar amplification in a multi-model ensemble is no doubt important for some applications but we do not think it is necessary to include it here. We prefer to keep our message simple and use this figure to show that polar amplification is robustly simulated in response to increases in $CO_2$.

10. Figure 2: 31 models are used in Figure 1 but only 25 in Figure 2. Why?

The figures were made by different groups using all of the available data at the time they were made.

11. Page 36, Lines 29-31: Maybe I am mixing things up, but, shouldn't this read this way: "For future, the lower (upper) quartile regression is used for SIC/SIT (SST), in order to give more weight to models with less sea ice and warmer SST. Conversely, for the pre-industrial period, the upper (lower) quartile is used ..."

We agree and have now corrected this – many thanks for spotting it.

---

## Author Comment (AC2) · 29 Aug 2018

Many thanks for your comments. Please see our replies below.

Anonymous Referee #2 This manuscript proposed several experiments for the PAMIP, one of the subset of CMIP6, in order to better understand the causes and consequences of polar amplification. I find this paper is well-written and only have some minor comments, as follows:

1. Caveat on AMIP simulations (tier 1 and tier 5) One interesting but controversial

[Figure]

**GMDD**

question in this area is whether observed recent Arctic sea ice loss since 1979 has significantly affected the jet stream and caused more winter cold extremes in the mid-latitudes (or the debate on "Warm Arctic, Cold Continents"). I am just wondering if PAMIP wants to touch on this question in tier 1. Tier 5 is very nice, but I am a little worried that the member size might be not enough to derive the forced sea ice loss effect for individual models. Sun et al. (2016) analyzed 50-member AMIP experiments to examine the "Warm Arctic, Cold Continents" hypothesis. Initially when they only had 10-member simulations, there was some cooling over Eurasia in response to Arctic sea ice loss. However, it turns out that is just internal variability because the cooling signal disappears when the ensemble size increases to 30. Similarly here, I suspect that 3 members might be not enough to show the forced response to Arctic sea ice loss for a single model. In other words, one should be able to use all available model ensemble to examine overall whether models show any response signal to observed sea ice loss but we may not be able to look at the sea ice loss effect in single models because of the large internal variability.

Reply: Understanding the impact of sea ice loss on the atmospheric circulation (and hence the "Warm Arctic Cold Continents" pattern) is indeed a key focus of PAMIP. This will be addressed in tier 1, but using idealised sea ice forcing from the differences between pre-industrial (pi), present day (pd) and future (fut) conditions. This choice (arrived at after much debate) provides multiple estimates (e.g. pd-pi, fut-pd, fut-pi) and is easily expanded to investigate additional aspects including the pattern of forcing and the roles of coupling and the background state. Improved understanding achieved through these experiments will provide much information on the role of sea ice in driving the Warm Arctic Cold Continents pattern. Experiment set 5 is optional, but provides additional focus on the recent period. We fully agree with the reviewer that a larger ensemble would be beneficial but we are wary of putting groups off by specifying experiments that are very costly, and take the view that any contributions would be welcome and would contribute to a larger multi-model ensemble. However, we have modified the text to highlight that larger ensembles are preferable if groups

have the resources to provide them.

2. Caveat on the coupled model experiments (tier 2 and tier 6). The design of tier 2 experiment appears to be different from most previous fully coupled ocean-atmosphere studies (Deser et al. 2015, 2016; Tomas et al. 2016; Smith et al. 2017; Oudar et al., 2017; Blackport and Kushner, 2017). From the point view of attribution study, I think the tier 6 more represents the full sea ice loss effect because in reality and in projected CMIP simulations the ocean will be allowed to justify when the sea ice declines (in other words sea ice reduction occurs gradually and ocean can adjust to that). Given the reason above, I feel that tier 2 can be used to demonstrate the role of quick ocean-atmosphere coupling but one has to look at tier 6 to understand the role of full ocean-atmosphere coupling (tier 2 might be more like slab-ocean coupled results to some extent).

Reply: We agree, and have noted this text.

Another small point: 100 years appear to be too short for tier 6. 200 years might be necessary because the first 100 years or so might need to be discarded due to model spin-up (ocean adjustment).

Reply: We agree that longer simulations would be beneficial and have added a note to encourage groups to provide them if possible.

3. Page 5 lines 5, 10: Liu et al. 2012 and Mori et al. 2014 are both modeling studies. Therefore, they probably should go to the next lines on "diverge considerable on the dynamical response", together with other studies that suggested recent Arctic sea ice loss did not significantly affect the jet stream and "Warm Arctic, Cold Continents" may simply reflect the atmospheric internal variability (e.g. Sun et al. 2016; Ogawa et al. 2018). These are comparable studies which all conducted AMIP-type simulations by prescribing recent Arctic sea ice loss.

Reply: We have included the additional references as suggested.

4. Page 7 line 10: isolating the response to sea ice (loss or change?). Also maybe I didn't fully understand the argument of "additional forcings". I thought we are discussing the effect of sea ice loss-alone. McCusker et al. (2017) have presented evidence that the atmospheric circulation response to sea ice loss and greenhouse gas forcing are remarkably linear. Thus it probably does not matter much.

Reply: We have clarified that additional steps are needed to isolate the impacts of sea ice in experiments that also include other forcings.

5. Page 9 Line 10: Blackport and Kushner, 2017

Reply: Corrected – thanks.

6. Figure 5b): there is a weird box in the bottom left

Reply: Corrected – thanks.

―――――――――――――――――――――――

---

## Author Response (AR2)

*Many thanks for your comments. Please see our replies in blue below.*

Topical Editor Decision: Publish subject to minor revisions (review by editor) (04 Sep 2018) by Julia Hargreaves

Comments to the Author:

Thanks for the revised manuscript. Referencing the line numbers for edits to the paper would have enabled a quicker response.

Nearly there I hope. A few comments.

1.

Reviewer:"9. Figure 1: How do we define polar amplification? as a mean of ratios or a ratio of means? 15 Hind et al., Scientific Reports (2016)."

Response:"How to quantify polar amplification in a multi-model ensemble is no doubt important for some applications but we do not think it is necessary to include it here. We prefer to keep our message simple and use this figure to show that polar amplification is robustly simulated in response to increases in CO2."

I think this is incorrect. There is no way of analysing an ensemble without adopting some method to do so. As someone who analyses ensembles, I'd very much like to know the most meaningful metric for the polar amplification community. I think what you have done for the Figure is actually something different from either of the two suggestions of the reviewer. The figure shows a mean of normalised fields, so if you took the polar to equator ratio it would be a ratio of normalised means...?

*We agree that Hind et al make some good points, and have replotted Figure 1 as a ratio of means. However, there is also diversity in the literature about whether polar amplification is defined by polar warming versus the global average, or versus the tropical average etc; as a 'ratio of trends' or a 'trend of a ratios'; which variable to use (surface temperature or something else); whether it is appropriate to calculate 'regional amplification' based on a limited longitude range, etc. We do attempt to resolve these issues in this paper, which is intended to document the protocol for multi-model experiments. The key point is that the model simulations are undertaken consistently so the community can effectively apply any metric and iterate towards a definition of best practice after further analysis.*

2.

The general idea is that a modeller, armed with a state of the art climate model and a nice big computer, should be able to set up and run the experiments using only the paper and permanently archived open access resources referenced in the paper. I think you are quite close but...

A. The Data request section is a bit difficult to understand. A modeller needs to know what the minimum set of requirements is. I think this might be the DCPP set plus Table 3, but I am not sure. Please clarify.

*Yes, you are correct, and we have clarified the data request.*

B. This is probably my stupidity, but I like to think that the aforementioned modeller should be allowed to be almost as thick as me. I could not find the boundary conditions for the experiments on input4mips. Either some more details need to be included in the appendix to help the modeller find what they need, or you need to upload the data to the database.

*The boundary condition fields are currently being tested and will be uploaded to the database as soon as possible. We have changed the first line in Appendix A to say that the data "will be" available from input4MIPs.*

---

## Author Response (AR3)

*Many thanks for your comments. Please see our replies in blue below.*

1. Your response to the question about the definition of how to define polar amplification, "there is also diversity in the literature about whether polar amplification is defined by polar warming versus the global average, or versus the tropical average etc; as a 'ratio of trends' or a 'trend of a ratios'; which variable to use (surface temperature or something else); whether it is appropriate to calculate 'regional amplification' based on a limited longitude range, etc. We do attempt to resolve these issues in this paper, which is intended to document the protocol for multi-model experiments. "

I am a bit confused, but assume you meant to write "do not" rather than "do" in the last sentence above...? If I understand that correctly, my response is that in a paper devoted to the comparison of polar amplification between models it is necessary to define what you are talking about. Therefore, please can you add a paragraph which discusses the different definitions, with references to the literature.

*Apologies, we did mean to write "do not", as you assumed. We have added some discussion in Section 5 to define polar amplification.*

2. Boundary conditions for model experiment description papers in GMD must be available before acceptance of the manuscript. If you are not able to get them on to input4mips in a reasonable timeframe, please upload them to an alternative public repository and provide a DOI for the upload in the revised version of the manuscript. (Ideally, a model experiment paper includes evidence of model output which demonstrate that the experiments work as expected. Thus the boundary conditions should actually have been finished - fully tested and ready to go - prior to submission of the manuscript!)

*We have made forcing data available at 10.5281/zenodo.1633416 and noted this in Section 9 of the manuscript.*

---

## Author Response (AR4)

Comments to the Author:

When I went to the Zenodo page for the boundary conditions data I saw, "You may request access to the files in this upload, provided that you fulfil the conditions below. The decision whether to grant/deny access is solely under the responsibility of the record owner."

*This wording is imposed by the Zenodo data platform.*

The data should be made fully publicly available. There is little point promoting an experiment for the climate community to use if the boundary forcing data is not available! Is there a reason why you cannot make the data available?

*The boundary forcing data are available! A restriction is imposed simply so that we can keep track of people who have downloaded this version of the data in case we need to make any further changes before making the data available on input4mips. We hope you agree that it would not be sensible to release the data without this safeguard.*

It is appreciated that you make the point that the version on input4mips may not be identical to the version discussed in the paper. However, rather than state "Preliminary forcing data for testing " ... are available on Zenodo, you should clearly state that the exact version of the boundary forcing discussed in the paper is the one that is available on Zenodo. Your wording suggests that you are not confident that the data are correct. This is concerning and I am inclined to hold the paper until the boundary conditions are finalised. It is generally required that experiments described in GMD papers should have already been performed by at least one model. Please can you confirm whether this is the case for your experiments.

*We have taken a lot of care to produce the forcing dataset (this has already delayed publication of the paper by several months) and do not expect it to change. The experiments have indeed been performed by two models. However, it is of course possible that an issue could arise before the official version of the data is released on input4mips. To further delay publication of the PAMIP protocol by holding the paper until the data are available on input4mips would be inconsistent and unfair given that many other CMIP6 protocol papers have already been published without this requirement.*

Whatever the answer to the above, it would be appropriate to create a version number for the current boundary conditions and to refer to this in the paper, and also on the Zenodo page. This is the norm for generic GMD experiment description papers, but it has not been enforced for all the CMIP6 experiments. However, as you seem to have a high expectation that the data will change, I think it would be a good idea here.

*The data has already been given a version number (version 2) on the Zenodo page. We now refer to this in the manuscript as requested. Furthermore, we have replaced*

*the original statement "Preliminary forcing data for testing are available" with "The latest version (version 2) of the forcing data, which we have presented here, is available", reflecting more accurately our expectation that the data are unlikely to change.*

---

## Author Response (AR5)

Comments to the Author:

Thank you for the revision. The paper is now very close to being acceptable. The only remaining issue is the accessibility of the data for the boundary conditions.

Despite being hosted on Zenodo, the data are restricted with access only being provided upon approval by the first author of the paper. I have exchanged a few emails with the first author and it appears that the authors are choosing not to make the data freely available, because they are concerned that it may be misused. Unfortunately this is not a valid reason to not make data available at GMD. We make our code and data open access in the same way that we make the scientific content of the paper open access, and the publication is incomplete if the code and data are not provided. Technically, there is a very simple solution to this problem and this is to remove the requirement for access to the data to be approved at Zenodo. At the same time, more information about the data should be added to the Zenodo page, as at present it is rather confusing and off putting. It seems to suggest that the data are restricted to those who fulfil certain criteria but does not state what those criteria are! It is fine for authors to keep a record of who accesses the data by requiring registration, but access should then be automatic. I tried to gain access myself, and the link I was sent did not work, and further human intervention was then required. I do not like such a system as I have quite a lot of experience with attempting to gain access to code or data that is supposedly available, but that requires human intervention. It is still too common in the community in general that requests for access are ignored.

I am happy to report, however, that I did finally gain access and that all the files were easily readable in netcdf format, and that the data look basically reasonable.

If the boundary conditions change before they are uploaded to the CMIP input4mips database, the authors should write a short update paper outlining the changes. They could also require that people running the experiments state in the data upload which version of the boundary conditions they used.

Note that every paper in GMD is assessed on its own merits against the peer review criteria, and not in reference to other papers already published. Special issue papers in GMD are reviewed in the same way, under the same criteria, and by the same editors as all other papers in the journal. Thus the authors should simply be aiming to publish an excellent GMD experiment description paper.

*It is critical that all groups contributing to CMIP6 use the same forcing data, otherwise it will be impossible to interpret the results. The official source of data for CMIP6 is via input4mips. Making interim versions freely available will potentially undermine the scientific credibility of CMIP6. Keeping an email log of those who have downloaded data is unsafe since it will require authors to continually check the log in the event of a change to the forcing data - there is no guarantee that this will happen.*

*For this reason we have now made the PAMIP data officially available from input4mips and removed the reference to Zenodo.*